# VFMF: Dense Forecasting by Generating Vision Foundation Model Features

**Gabrijel Boduljak** [1]   **Yushi Lan** [1]   **Christian Rupprecht** [1]   **Andrea Vedaldi** [1]

Project Page | Code

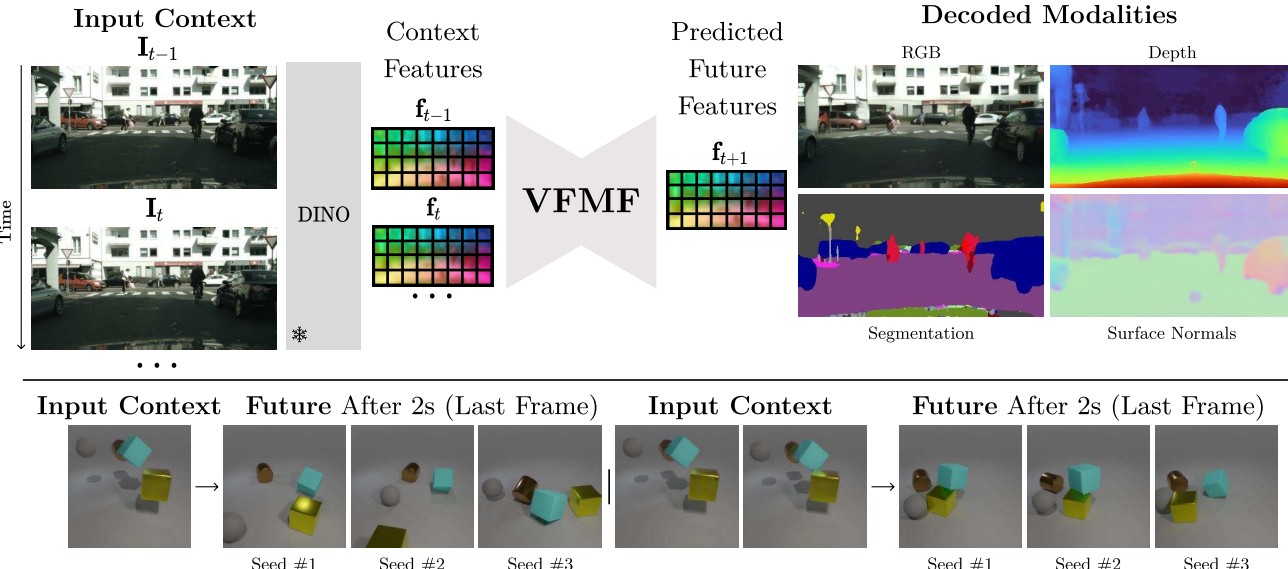

*Figure 1.* **An overview of our method.** Our scene forecasting method, **VFMF**, *autoregressively generates* diverse futures in the *latent space* of a *foundation model*, translatable into downstream modalities, such as segmentation, depth, surface normals and even RGB.

## Abstract

Forecasting by generating RGB videos is computationally expensive, often physically implausible, and not directly actionable, since it requires translation into decision-making signals. Direct modality forecasting (e.g., predicting future segmentation) produces directly actionable outputs but fails to scale due to the need for labels. Vision Foundation Model (VFM) features offer the best of both worlds: they contain actionable semantic and geometric information that can be easily decoded from the predicted features, while requiring no labels on the downstream task for training. However, almost all existing VFM feature forecasting methods regress future features from fixed number of input frames, with evaluation predominantly on short horizons matching the training

setup. We firstly show that existing regression methods struggle with forecasting from partial observations because they average over multiple plausible futures, failing to capture uncertainty in the future given the past. Interestingly, naively replacing deterministic forecasting with generative flow matching does not match the sample quality of the regression model, despite being a mathematically appropriate formulation of the forecasting task. In this work, we explain why this is the case, and we show how to optimally generate foundation model features. Our key insight is that generative modeling of VFM features requires (auto)encoding into a compact latent space suitable for diffusion. We show that this latent space preserves information more effectively than previously used alternatives, such as uncompressed feature diffusion or PCA-based compression, both for forecasting and other applications, such as image generation. Our results suggest that conditional generation of (compressed) VFM features offers a promising and scalable foundation for future scene forecasters.

[1]Visual Geometry Group, University of Oxford. Correspondence to: Gabrijel Boduljak <gabrijel@robots.ox.ac.uk>.

*Proceedings of the $43^{rd}$ International Conference on Machine Learning*, Seoul, South Korea. PMLR 306, 2026. Copyright 2026 by the author(s).

# 1. Introduction

One of the key challenges in scene forecasting is to predict future states of a scene from partial observations of its past (Ha & Schmidhuber, 2018). Yet, there is significant debate on how to tackle this problem, starting with the choice of scene representation. Many have suggested that forecasters could be video generators (Brooks et al., 2024; NVIDIA et al., 2025; Bruce et al., 2024). These are appealing for three reasons. First, when implemented incrementally or autoregressively, they effectively predict future pixels from past ones. Second, because they process pixels, they can be trained on enormous amounts of video data with minimal curation and manual supervision. Third, the quality of these models has improved dramatically in recent years.

A key downside of pixel-based models, however, is that predicting pixels is neither necessary nor directly useful in many applications. Predicting pixels is computationally intensive, and generators often make physically implausible predictions (Kang et al., 2025; Boduljak et al., 2026; Motamed et al., 2025). The output of forecasters should be actionable and serve as the basis for decision-making. For example, we may decide to steer a car if failing to do so is predicted to cause an accident. However, generating an *image* of an accident is not the same as understanding that an accident may occur. The latter requires parsing the image to extract its meaning. By generating pixels, we may be doing unnecessary work and producing outputs that are needlessly complex to interpret.

Thus, even before video generators became popular, authors considered alternative, more useful, and compact representations for scene forecasting (Luc et al., 2017; 2018). A particularly interesting approach is based on the representations computed by vision foundation models (VFMs), such as DINO (Caron et al., 2021; Oquab et al., 2024; Siméoni et al., 2025). VFM feature spaces tend to capture information that is more easily decoded into semantic properties (e.g., object classes) that are directly actionable. In fact, they are semi-dense and can also be decoded into geometric properties such as depth maps, which may be more directly useful to an agent operating in the physical world. Furthermore, they may disregard low-level image details that are irrelevant to the task at hand, whose modeling can be wasteful.

Consequently, several authors (Karypidis et al., 2026; Zhou et al., 2025; Baldassarre et al., 2025; Walker et al., 2025) have suggested basing scene forecasting on these feature spaces. However, unlike video generators, which are inherently stochastic and can model uncertainty about the future, these works have so far focused only on *deterministic forecasting*. In particular, none of these works considers prediction from variable number of input frames, while this is a standard practice in video generation. Furthermore, in all these works, the number of input frames during evaluation is fixed, matching the training number. This evaluation does not assess model's ability to capture uncertainty about future given the past, which we argue is crucial for real scene forecasting, because the real-world is almost always partially observed. Thus, in this work, we consider forecasting from variable number of input frames, while keeping the setup as close as possible to prior works for fair comparison. Here, as is well known, and as we further show in our experiments, deterministic prediction of ambiguous targets yields blurry predictions that compound during rollout (Figure 3), demonstrating that existing deterministic feature forecasting methods are ill-suited for this problem. Next, perhaps surprisingly, we show that naively replacing regression with flow matching does not match sample quality of the regression model (Figure 5), assuming the same architecture and training data, although it is mathematically correct way to formulate our task. This motivates the following question:

*How can we build generative forecasters in the feature spaces of foundation models?*

To answer this question, we start by noting that effective diffusion latents are typically low-dimensional and trained with specialized variational autoencoders (VAEs) (Kingma & Welling, 2014), whereas features extracted by VFMs are often high-dimensional, making diffusion ill-conditioned and unstable (Hoogeboom et al., 2025). One solution is principal component analysis (PCA)–based compression, and this has been used in deterministic VFM forecasting (*DINO-Foresight* (Karypidis et al., 2026)) or image generation (*ReDi* (Kouzelis et al., 2025b)). We argue that this is not very effective due to information loss from PCA (Figure 5).

Additionally, we perform *spectral decomposition* of DINO features on several datasets, showing that *spectral properties* of DINO features follow those of RGB. Thus, motivated by recent works on relationship between *spectral properties* of diffusion space and generation quality (Skorokhodov et al., 2025; Kouzelis et al., 2025a; Falck et al., 2025), we propose learning a *compact latent space on top of VFM features* using a new VAE for this purpose. We show that our VAE-compressed VFM feature space yields a compact, well-conditioned latent space that better preserves semantic and geometric information. Within a latent diffusion model, it enables uncertainty-aware, temporally coherent forecasting that remains robust across different context lengths.

To summarize, our contributions are as follows: (i) We show that deterministic, regression-based forecasters perform poorly with variable and short context: they collapse multimodal[1] predictions to means that are not necessarily meaningful. (ii) We use *autoregressive flow matching* for forecasting in VFM feature spaces, yielding uncertainty-

---
[1]In a distributional sense.

aware, coherent predictions that work well with different context lengths. (iii) We introduce a *VAE-compressed VFM feature space* that preserves useful information much better than the PCA compression used in prior work. (iv) We demonstrate significant improvements in the forecasting of semantic and geometric quantities, as well as RGB.

## 2. Related Work

**Vision Foundation Models.** Vision foundation models (VFMs) have transformed visual representation learning by training on large-scale image, video, and multimodal datasets. Self-supervised approaches, such as DINO and DINOv2 (Caron et al., 2021; Oquab et al., 2024; Siméoni et al., 2025), learn rich visual embeddings through self-distillation. In contrast, vision-language models like CLIP (Radford et al., 2021; Fang et al., 2024) align visual and textual spaces, while SAM (Kirillov et al., 2023) enables open-set segmentation. Video and 3D extensions, including VideoMAE (Assran et al., 2025; Carreira et al., 2025) and VGGT-like models (Wang et al., 2025), extend these representations across space and time. Although primarily designed for perception, recent work shows that pretrained VFM features can also benefit generation and reconstruction tasks (Zheng et al., 2026; Kouzelis et al., 2025b). Building on this insight, we explore how the latent spaces of VFMs can serve as compact, semantically meaningful basis for generative forecasting.

**Scene Forecasting in Feature Space.** Vondrick et al. 2016 reformulate this task as *future frame regression* in the latent space of pre-trained, supervised feature extractor, such as AlexNet (Krizhevsky et al., 2012). More recent works, such as *DINO-Foresight* (Karypidis et al., 2026), *DINO-WM* (Zhou et al., 2025), and *DINO-World* (Baldassarre et al., 2025) extend this line of work by replacing supervised representations with self-supervised DINO. By operating in semantically structured latent spaces, these methods achieve more stable and interpretable forecasting than pixel-based alternatives (e.g., VISTA (Gao et al., 2024b)), enabling downstream tasks such as semantic and geometric prediction (Luc et al., 2018; Karypidis et al., 2026). However, these models are regression-based and deterministic, leading to *averaged* and unrealistic predictions under multimodal or uncertain futures. Our work addresses this limitation by introducing a generative formulation that explicitly models uncertainty with conditional distributions in the VFM space, compressed by a VAE. Similarly to our work, *Generalist Forecasting* also propose a generative forecasting method, but on uncompressed VFM space, with fixed context length.

**Vision Foundation Model Features For Generation.** Several recent works such as RAE (Zheng et al., 2026) and REPA (Yu et al., 2025) have explored the integration of pretrained feature representations into diffusion models.

RAE decomposes image generation into feature generation followed by RGB decoding. To handle high-dimensional diffusion, it widens the denoising network to match feature dimension, which is computationally impractical in our setting. Instead, we use standard latent compression and VAE regularization in the novel setting of feature forecasting, allowing us to preserve conventional denoising architectures and hidden dimensions. Importantly, RAE generates only final-layer DINO features for a single image, whereas our method predicts multi-scale DINO features from four layers, conditioned on several context frames represented as patch-level features. A naive extension of RAE to our setting would already require a transformer hidden dimension of at least 3072 for DINO-Base features alone, before accounting for temporal context, making it infeasible on our hardware. In contrast, both our VAE and denoising transformer operate at a hidden dimension of 768, consistent with prior work such as *DINO-Foresight*.

REPA also targets image generation, but uses pretrained encoder features as supervision to align the representations of an RGB latent diffusion model to accelerate training convergence. In contrast, our method generates features directly as the output. Moreover, our baseline, ReDi, already outperforms REPA on image generation, and we further improve upon ReDi by replacing its PCA-based DINO latents with our VAE-based DINO latents.

In summary, our work differs from both methods in its problem setting. While REPA and RAE focus exclusively on improving pixel-level image generation quality and convergence, we tackle the distinct problem of forecasting future representations from a variable number of context frames. Beyond problem setting, we also provide insights absent from both previous works: we show that a VAE specifically improves temporal consistency and shape geometry compared to alternative compression approaches (Figure 5), whereas REPA and RAE report image generation metrics without identifying which aspects of generation improve.

## 3. Method

We design a model that generates future states of a scene represented in the latent space of vision foundation models (VFMs). Unlike prior works (Karypidis et al., 2026; Zhou et al., 2025; Baldassarre et al., 2025), we generate future VFM features conditioned on a *variable-length* context of past observations. An overview of our method is in Figure 2.

### 3.1. Background: Deterministic VFM forecasting

We begin by describing the idea of deterministic forecasting in VFM feature spaces recently revisited by Karypidis et al.; Zhou et al.; Baldassarre et al..

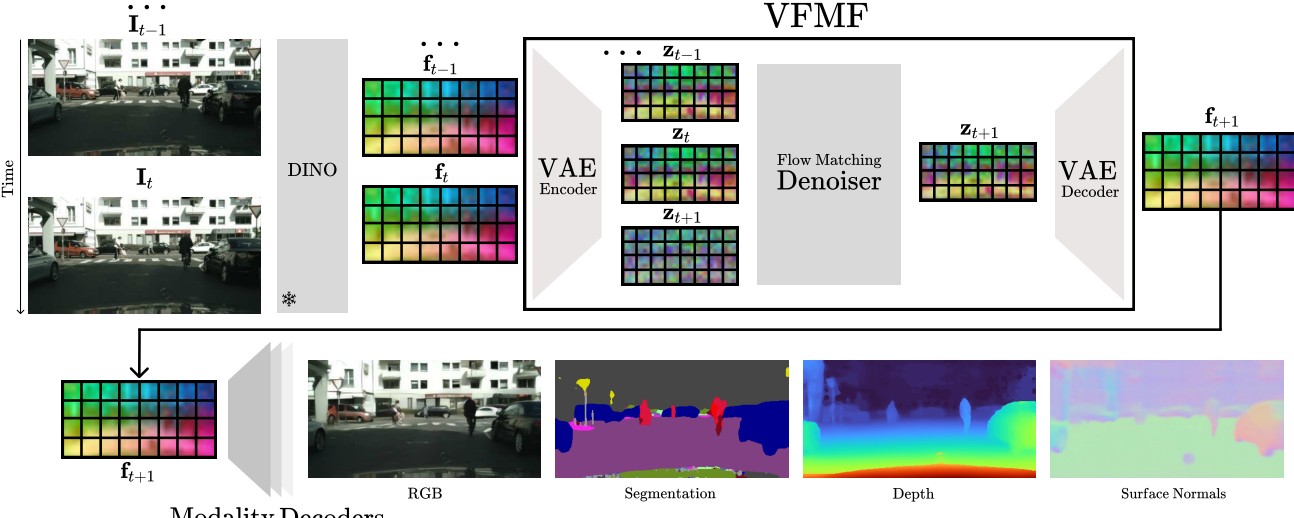

*Figure 2.* **An overview of our method VFMF.** Given RGB context frames $\mathbf{I}_1, \ldots, \mathbf{I}_t$, we extract DINO features $\mathbf{f}_1, \ldots, \mathbf{f}_t$ and predict the next state feature $\mathbf{f}_{t+1}$. Context features are compressed with a VAE along the channel dimension to produce context latents $\mathbf{z}_1, \ldots, \mathbf{z}_t$. Those context latents are concatenated with noisy future latents $\mathbf{z}_{t+1}$ and passed to a conditional denoiser that denoises only the future latents $\mathbf{z}_{t+1}$ while leaving the context latents unchanged. This process repeats autoregressively, with a window of fixed length. Specifically, each time a new latent $\mathbf{z}_{t+1}$ is generated, it is appended to the context while the oldest context latent is popped. The denoised future latents are decoded back to DINO feature space by the VAE decoder. Finally, the reconstructed features can be routed to task-specific modality decoders for downstream tasks or interpretation.

Let a video be a sequence $\{(\boldsymbol{v}_t, t)\}_{t=1}^{T}$ of $t$ frames, where the tensor $\boldsymbol{v}_t \in \mathbb{R}^{H' \times W' \times 3}$ is a video frame. A VFM encoder (e.g., DINOv2 (Oquab et al., 2024)) maps each frame to features $\boldsymbol{f}_t = \text{ENCODER}(\boldsymbol{v}_t) \in \mathbb{R}^{H \times W \times D}$. Given a context $\boldsymbol{f}_{1:T}$ of length $T$, regression-based methods predict future features $\boldsymbol{f}_{t'}$ for $t' > T$ and, if trained to minimize the $\ell_2$ loss, approximate the conditional mean $\mathbb{E}[\boldsymbol{f}_{t'} \mid \boldsymbol{f}_{1:t}]$. In general, but in particular when $T$ is small (e.g., 1–2 frames), the state of the scene is underdetermined since *many futures are plausible*. A single point estimate averages these hypotheses, yielding over-smoothed predictions that become worse as the context shortens and rollouts become longer (Figure 3).

### 3.2. Generative (Stochastic) VFM forecasting

In order to address the limitations of deterministic forecasting, we propose a *generative* model that captures uncertainty over future features.

Formally, this amounts to learning a conditional distribution $p(\boldsymbol{f}_{T+1} \mid \boldsymbol{f}_{1:T})$ from which several plausible futures can be sampled. As the context ($\mathcal{C}$) length $T$ grows, predictive uncertainty naturally decreases and forecasts sharpen; with shorter context, sample variability reflects the increasing ambiguity rather than collapsing to an average. Conceptually, this mirrors autoregressive video generation, but we generate *VFM features* rather than pixels[2].

We implement this with *latent flow matching* (Lipman et al., 2023) on a compact feature latent (Section 3.2.1).

### 3.2.1. AUTO-ENCODING VFM FEATURES

A challenge of forecasting in VFM feature space is that these features are high-dimensional, with $D$ in the hundreds or thousands of feature channels. Prior works have suggested to PCA-compress these features (Karypidis et al., 2026; Kouzelis et al., 2025b), but this discards too much information, harming generation quality.

We thus propose to train a VAE (Kingma & Welling, 2014) over the VFM features to obtain a generative-friendly compact latent code $\boldsymbol{z} \in \mathbb{R}^{H \times W \times D/r}$. The goal of the VAE is to reduce the feature dimension by a large factor $r$ (whereas the spatial dimension is preserved as the VFM features are already spatially downsampled compared to the input images). Based on the VAE formulation, the encoder outputs a diagonal Gaussian posterior with parameters $(\boldsymbol{\mu_z}, \boldsymbol{\sigma_z}) = \boldsymbol{\phi}(\boldsymbol{f})$, and the decoder reconstructs $\hat{\boldsymbol{f}} = \psi(\boldsymbol{z})$ from a sampled latent $\boldsymbol{z} \sim \mathcal{N}_{(\boldsymbol{\mu_z}, \boldsymbol{\sigma_z})}$. We train by optimizing the $\beta$-VAE loss

$$\mathcal{L}_{\beta\text{-VAE}} = \mathbb{E}_{\boldsymbol{z} \sim \mathcal{N}_{\boldsymbol{\phi}(\boldsymbol{f})}}[\tfrac{1}{2}\|\boldsymbol{f} - \psi(\boldsymbol{z})\|_2^2] + \beta \cdot \mathbb{D}_{\text{KL}}(\mathcal{N}_{\boldsymbol{\phi}(\boldsymbol{f})}\|\mathcal{N}_0)$$

where $\mathcal{N}_0$ is standard normal. Note that frames are auto-encoded independently.

---

[2]Despite the intent of VAEs to learn abstract representations, the latents produced by state-of-the-art RGB VAEs remain rela-

tively low-level: they inherit the spatial grid structure of the input, closely resembling a downsampled version of the input image or video (Dieleman, 2025).

## 3.3. Forecasting using Rectified Flow

Having mapped the VFM features $f_t$ to compact latents $z_t$ with the VAE, we now learn a stochastic forecasting model in this latent space. Namely, instead of learning the distribution $p(f_{T+1} \mid f_{1:T})$, we learn $p(z_{T+1} \mid z_{1:T})$.

We do so by using *rectified flow/flow matching* (Liu et al., 2023; Lipman et al., 2023). A velocity network $\hat{v}_\theta(z^{(t)}, \mathcal{C}, t)$ is trained with the standard objective.

During training, we *randomize* the length of the context ($|\mathcal{C}| = T$) in the range 1 to $K$, so the model calibrates uncertainty to the available history, for example, learning that shorter contexts result in increased ambiguity. Following prior works, $K = 4$.

At test time we sample $z^{(0)} \sim \mathcal{N}(0, I)$ and integrate the learned ODE $\dot{z}^{(t)} = \hat{v}_\theta(z^{(t)}, \mathcal{C}, t)$ from $t = 0$ to $1$ to obtain $\hat{z}_{T+1} = z^{(1)}$. We roll out autoregressively with a sliding window of length $K$, i.e.,

$$p(z_{T+1} \mid z_{1:T}) \approx p(z_{T+1} \mid z_{T-K+1:T}).$$

## 3.4. Decoding Multiple Modalities

Forecasting in VFM feature space simplifies decoding the prediction to several useful interpretable modalities. For semantic segmentation, depth, and surface normals, we follow *DINO-Foresight* (Karypidis et al., 2026) or *Generalist Forecasting* (Walker et al., 2025) and use simple regression or classification heads. For RGB reconstruction, we either use a ViT-B (Dosovitskiy et al., 2020) backbone with a DPT-based decoder (Ranftl et al., 2021) or attention-based readout (Carreira et al., 2025), trained with LPIPS (Zhang et al., 2018) and $\ell_1$ or $\ell_2$.

**Discussion.** While it is difficult to assess the benefits of a VAE directly in VFM space, we can instead measure its effect on downstream decoding tasks. Here we discuss one such example and investigate others in Section 4. It is well known that many high-bandwidth visual features are approximately *invertible*, in the sense that the input image can be reconstructed from them (Mahendran & Vedaldi, 2015; Nguyen et al., 2016). We therefore train an *inverter* (i.e., a network that maps features back to an image) and evaluate whether inversion remains possible after compressing and decompressing the features. Figures 3 and 5 shows that VAE compression substantially outperforms PCA, which yields blurry reconstructions and information loss.

## 4. Experiments

In Section 4.1, we begin by demonstrating that stochastic VFM forecasting performs better than deterministic regression, emphasizing the importance of explicitly modeling uncertainty in world modeling. In Section 4.2, we exten-

sively analyze the effect of different diffusion spaces on the sample quality, justifying the importance of optimal autoencoding of VFM features. Finally, in Section 4.3, we show that VFM auto-encoding is preferable to PCA compression not only in forecasting, but also in image generation. Specifically, we use it to improve *ReDi* (Kouzelis et al., 2025b) the state-of-the-art joint image and feature generator on *ImageNet* (Deng et al., 2009).

## 4.1. Scene Forecasting

VFMF forecasts features, which we then decode into various modalities such as semantic segmentation, depth, and surface normals. We assess the quality of these predictions.

We use DINOv2 (Oquab et al., 2024) as our VFM to enable direct comparison with *DINO-Foresight* (Karypidis et al., 2026), a deterministic regression baseline, and *Generalist Forecasting* (Walker et al., 2025), a generative baseline. Since neither baseline handles variable context lengths and only *DINO-Foresight* is publicly available, we retrained it with variable-length contexts while keeping other settings identical. For *Generalist Forecasting*, whose code is unavailable, we trained our method with fixed context matching their protocol. *DINO-Foresight* serves as a representative regression method since *DINO-World* (Baldassarre et al., 2025), another regression approach, has no available implementation and performs comparably on *Cityscapes*.

**Datasets.** We evaluate on *Cityscapes* (Cordts et al., 2016), *Kubric MOVi-A* (Greff et al., 2022) and *ScanNet* (Dai et al., 2017). These are complementary: *Cityscapes* and *ScanNet* offer diverse real-world dynamics but a single annotated future per clip, whereas *Kubric* enables controlled generation of multiple plausible futures under uncertainty.

**Benchmark.** We evaluate *future forecasting* using several modalities: semantic segmentation, depth, surface normals and RGB. For comparison with *DINO-Foresight*, we use DPT (Ranftl et al., 2021) decoding heads that map predicted VFM features to targets. On *Cityscapes*, we use the official probing heads released by *DINO-Foresight* for their model; for ours, we train new heads under the same protocol and codebase. On *Kubric*, we train new probing heads for all methods within the shared implementation framework. For comparison with *Generalist Forecasting* (Walker et al., 2025), we train attention-based probes following their protocol, since the official implementation is not available.

It is important to note that our evaluation is not the same as in prior works, such as *DINO-Foresight*. These evaluate forecasting from the fixed number of frames, while we evaluate from variable number of input frames. Additionally, our temporal horizon is *3x* longer than *MID* in prior works.

**Metrics.** Following *DINO-Foresight*, we report semantic segmentation with mIoU over all classes (mIoU-All) and over movable objects only (MO-mIoU); depth with AbsRel and $\delta_1$; and surface normals with mean angular error $m$ and the percentage of pixels with error $< 11.25°$. On *Kubric*, because the number of instances varies per scene, we evaluate foreground/background segmentation only.

To compare fairly with deterministic regression baseline *DINO-Foresight*, we use two evaluation protocols for our stochastic model:

- **Mean-of-$k$**: average $k$ sampled futures *in feature space* to obtain a single prediction (an estimate of the conditional mean given the context), then decode once. This matches the $\ell_2$ regression target.
- **Best-of-$k$**: compute the metric for each of the $k$ samples and report the best score. On *Cityscapes* there is one ground truth per clip; on *Kubric*, single-frame rollouts have 64 ground-truth futures per scene, whereas rollouts from $\geq 2$ frames are effectively deterministic (horizontal velocities are observable), so a single ground truth is used.

We use $k=32$ on *Cityscapes* and $k=64$ on *Kubric* to balance computational efficiency and evaluation accuracy. Results for different $k$ are in Tables 8 to 11.

To compare fairly with generative baseline *Generalist Forecasting*, we follow their evaluation protocol. We sample from each method $k = 10$ times per example and report average and best results, along with Frechet distance in RGB or depth space. Here, both methods take 4 frames as fixed context and forecast the next 12 frames.

More details about the implementation, ablations, datasets and evaluation protocol are provided in Sections A, C, D.1 and D.2, respectively.

**Results.** Table 1 shows that our generative method substantially outperforms the deterministic baseline on both real and synthetic data.

The performance gap is largest when uncertainty is highest (i.e., at shorter context lengths), directly validating our key contribution: *explicit uncertainty modeling*. While all methods benefit from longer context lengths, our approach consistently achieves superior performance across all evaluation settings by dynamically adapting to the provided information from the input frames (Figure 12).

To isolate the source of our improvements, we trained the baseline with our VAE latents (*DINO-Foresight (V)*) instead of high-rank PCA projections of DINO features. This baseline performs comparably to, or slightly better than the PCA version, yet remains substantially worse than both our mean and best predictions. Since our denoising architecture is

derived from *DINO-Foresight* with minimal modifications (Section C.3), our gains indeed stem from uncertainty modeling itself, not from a superior representation space.

As shown in Table 2, our generative method substantially outperforms *Generalist Forecasting*, the generative baseline that diffuses features directly. This underscores the critical advantage of VAE compression: unlike the baseline, which generates features directly, our results demonstrate that diffusing compact latent features is key to superior generation.

*Table 1.* **Dense forecasting accuracy.** Our *generative* VFMF considerably outperforms *deterministic DINO-Foresight* on both *Cityscapes* and *Kubric*, highlighting benefits of explicit modeling of uncertainty in future.

| Model | Segmentation | | Depth | | Normals | |
|---|---|---|---|---|---|---|
| | All (↑) | Mov. (↑) | d1 (↑) | AbsRel (↓) | a3 (↑) | MeanAE (↓) |
| *initial context length* $\vert\mathcal{C}\vert = 1$ | | | | | | |
| **Cityscapes** (*evaluation at 18th future frame after 9 autoregressive rollouts*) | | | | | | |
| *DINO-Foresight* | 31.67 | 22.00 | 70.77 | 0.23 | 84.82 | 5.33 |
| *DINO-Foresight* (V) | 30.62 | 20.04 | 71.65 | 0.23 | 84.96 | 5.28 |
| VFMF (Mean) | 34.85 | 31.76 | 75.89 | 0.19 | 88.48 | 4.55 |
| VFMF (Best) | **41.69** | **38.93** | **80.43** | **0.16** | **90.66** | **4.18** |
| **Kubric** (*evaluation at 22th future frame after 11 autoregressive rollouts*) | | | | | | |
| *DINO-Foresight* | 46.73 | 5.11 | 64.37 | 0.24 | 90.62 | 2.94 |
| *DINO-Foresight* (V) | 47.10 | 5.20 | 61.93 | 0.25 | 91.16 | 2.78 |
| VFMF (Mean) | 48.29 | 6.61 | 68.33 | 0.21 | 93.29 | 2.14 |
| VFMF (Best) | **70.55** | **47.77** | **88.80** | **0.08** | **93.45** | **2.07** |
| *initial context length* $\vert\mathcal{C}\vert = 2$ | | | | | | |
| **Cityscapes** (*evaluation at 16th future frame after 8 autoregressive rollouts*) | | | | | | |
| *DINO-Foresight* | 39.35 | 32.26 | 74.34 | 0.21 | 87.14 | 4.86 |
| *DINO-Foresight* (V) | 40.13 | 31.92 | 77.14 | 0.19 | 88.90 | 4.50 |
| VFMF (Mean) | 41.74 | 38.96 | 78.63 | 0.17 | 90.87 | 4.09 |
| VFMF (Best) | **45.23** | **42.60** | **81.93** | **0.15** | **92.14** | **3.88** |
| **Kubric** (*evaluation at 20th future frame after 10 autoregressive rollouts*) | | | | | | |
| *DINO-Foresight* | 51.15 | 14.39 | 62.60 | 0.24 | 89.20 | 3.36 |
| *DINO-Foresight* (V) | 51.85 | 15.24 | 64.69 | 0.24 | 89.64 | 3.23 |
| VFMF (Mean) | 55.89 | 21.68 | 68.87 | 0.22 | **92.31** | 2.39 |
| VFMF (Best) | **64.84** | **37.97** | **78.06** | **0.16** | 91.66 | **2.59** |
| *initial context length* $\vert\mathcal{C}\vert = 3$ | | | | | | |
| **Cityscapes** (*evaluation at 14th future frame after 7 autoregressive rollouts*) | | | | | | |
| *DINO-Foresight* | 41.89 | 35.61 | 75.91 | 0.19 | 88.42 | 4.60 |
| *DINO-Foresight* (V) | 44.48 | 38.21 | 79.14 | 0.17 | 90.71 | 4.16 |
| VFMF (Mean) | 44.53 | 41.87 | 79.72 | 0.16 | 91.78 | 3.92 |
| VFMF (Best) | **47.39** | **44.84** | **82.72** | **0.15** | **92.88** | **3.73** |
| **Kubric** (*evaluation at 18th future frame after 9 autoregressive rollouts*) | | | | | | |
| *DINO-Foresight* | 54.28 | 19.95 | 66.76 | 0.22 | 89.43 | 3.26 |
| *DINO-Foresight* (V) | 54.96 | 20.90 | 68.73 | 0.22 | 89.70 | 3.17 |
| VFMF (Mean) | 59.47 | 27.79 | 71.93 | 0.21 | **92.67** | 2.26 |
| VFMF (Best) | **68.25** | **43.74** | **80.09** | **0.14** | 92.31 | **2.40** |
| *initial context length* $\vert\mathcal{C}\vert = 4$ | | | | | | |
| **Cityscapes** (*evaluation at 12th future frame after 6 autoregressive rollouts*) | | | | | | |
| *DINO-Foresight* | 44.75 | 38.92 | 77.66 | 0.18 | 89.87 | 4.31 |
| *DINO-Foresight* (V) | 47.17 | 40.94 | 80.51 | 0.16 | 92.05 | 3.89 |
| VFMF (Mean) | 46.49 | 43.90 | 80.72 | 0.16 | 92.53 | 3.77 |
| VFMF (Best) | **49.43** | **46.97** | **83.56** | **0.14** | **93.59** | **3.59** |
| **Kubric** (*evaluation at 16th future frame after 8 autoregressive rollouts*) | | | | | | |
| *DINO-Foresight* | 57.62 | 25.78 | 69.31 | 0.22 | 89.82 | 3.11 |
| *DINO-Foresight* (V) | 58.03 | 26.29 | 69.15 | 0.23 | 90.11 | 3.04 |
| VFMF (Mean) | 61.74 | 31.86 | 71.88 | 0.21 | **92.66** | **2.25** |
| VFMF (Best) | **69.86** | **46.56** | **80.50** | **0.14** | 92.62 | 2.31 |

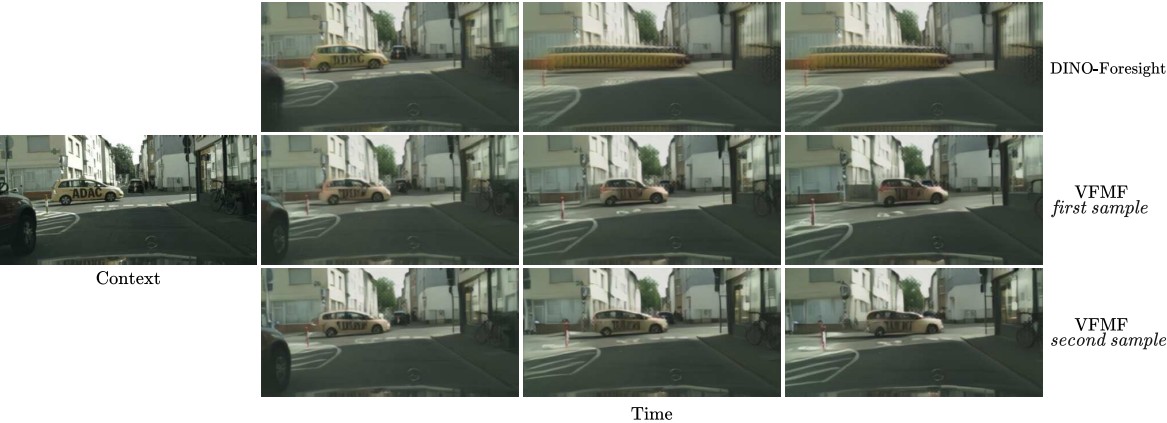

*Figure 3.* **Qualitative comparison of future predictions** translated into RGB domain. DINO-Foresight, a *regression* baseline, is unable to model uncertainty in the motion of both the ego-vehicle and the car in the middle of the street, effectively averages all possible futures, producing blurry and physically implausible predictions. In contrast, our *generative* method generates plausible futures, accurately capturing the uncertainty in unknown velocities and accelerations, that can be translated into sharp RGB or other modalities (Figure 4).

*Table 2.* **Comparison with *Generalist Forecasting* (DINOv2).** We report RGB and Depth metrics, where our method (VFMF) consistently outperforms the baseline (*Generalist Forecasting*).

|  | FD ($\downarrow$) | Mean | Std | Best |
|---|---|---|---|---|
| *RGB* (PSNR $\uparrow$) |  |  |  |  |
| *Generalist Forecasting* | 82.30 | 15.28 | 0.53 | 16.14 |
| VFMF | **14.08** | **17.64** | 2.39 | **19.53** |
|  |  |  |  |  |
| *Depth* (AbsRel $\downarrow$) |  |  |  |  |
| *Generalist Forecasting* | 626.75 | 0.15 | 0.05 | **0.08** |
| VFMF | **62.88** | **0.09** | **0.04** | **0.08** |

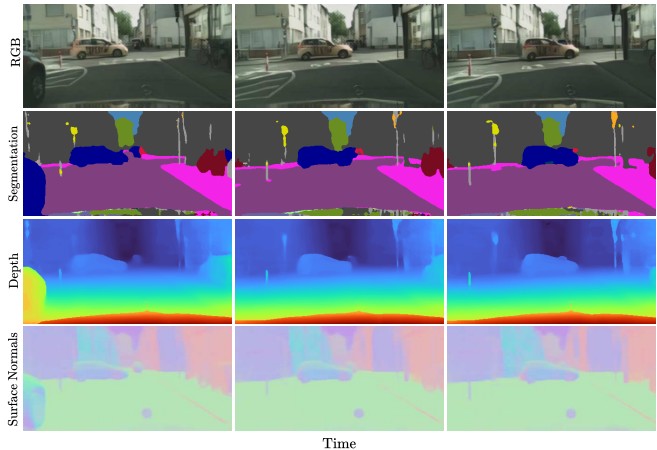

*Figure 4.* **One feature set, many modalities:** Our diverse generated futures (Figure 3) can be translated to diverse modalities, from pixels (RGB) to semantics and geometry (depth, normals).

## 4.2. How (Not) to Diffuse DINO Features?

Once we frame scene forecasting as conditional generation, a straightforward approach is to diffuse DINO features directly. However, as shown in Figure 5, this produces unsatisfactory results even on the simple *Kubric* dataset. Specifically, direct diffusion of DINO features leads to unrealistic motion with several failure modes: distorted object geometry, objects merging together, and non-rigid deformations. When the diffused features are translated to downstream modalities, such as RGB, they also exhibit various artefacts, indicating residual noise resulting from imperfect denoising.

**Feature Compression Is Crucial for Generation Quality.** First, we hypothesize that the aforementioned errors are *primarily* a consequence of the curse of dimensionality - *diffusing thousands of channels, while requiring temporal consistency is too difficult*. In fact, modern image/video diffusion models (Chen et al., 2025b; Peng et al., 2025; Wan et al., 2025; Sand.ai et al., 2025; HaCohen et al., 2024) diffuse *heavily compressed* RGB latents, by up to 192×. To test this hypothesis, we compress features (by 192×) with two methods, namely PCA and VAE, while keeping the

latent dimension fixed. Figure 5 shows that diffusing 16 channels significantly outperforms diffusing DINO features directly or PCA-compressed with a higher rank (1152 instead of 16). In particular, most of the errors in geometry or shape inconsistency through time disappear, as further evidenced by sharply decoded geometry (surface normals, depth). This also reflects in significantly improved downstream forecasting accuracy on both datasets (Section A). This strongly supports the hypothesis that low-dimensional latent diffusion is easier than direct feature diffusion.

**PCA Compression Is Suboptimal For Generation.** Although low-rank PCA simplifies latent diffusion, it incurs noticeable information loss. Specifically, Figure 5 shows that while high-level properties such as semantics and geometry (normals) are well-preserved, pixel-level details are lost.

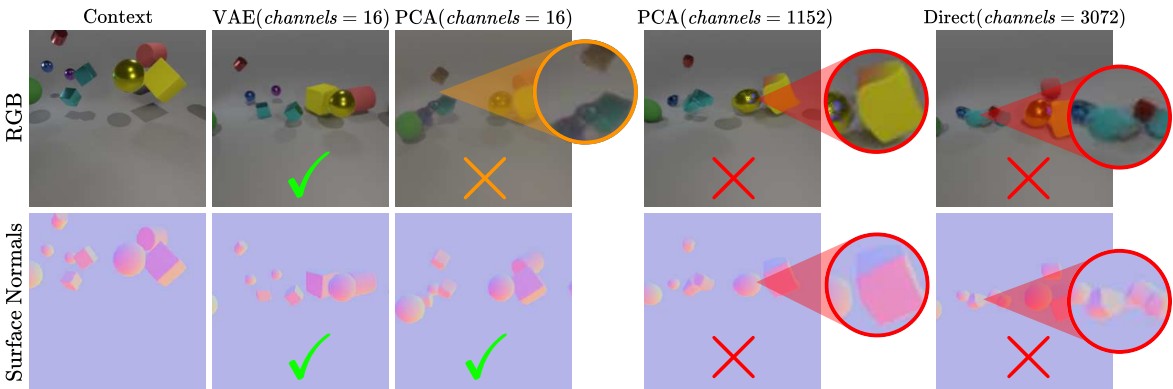

*Figure 5.* **Qualitative comparison** of future prediction decoded from different latent spaces. Diffusing VAE latents significantly outperforms diffusing PCA-compressed latents or features directly, producing more geometrically consistent predictions with higher-quality RGB output. Alternative methods exhibit two several failure modes: inconsistent geometry, shape distortions and loss of RGB details. Low-rank PCA maintains geometric consistency but loses fine-grained RGB details. Increasing the PCA rank improves RGB fidelity but degrades shape consistency. Direct diffusion and high-rank PCA both suffer from geometric distortions and RGB artefacts.

*Table 3.* **Dense decoding performance of decoding heads**, evaluated on ground truth annotated futures. Under equal latent capacity, PCA exhibits severe degradation, while the VAE maintains high reconstruction quality, significantly higher than PCA.

| Model | Segmentation | | Depth | | Normals | |
|---|---|---|---|---|---|---|
| | All (↑) | Mov. (↑) | d1 (↑) | AbsRel (↓) | a3 (↑) | MeanAE (↓) |
| **Cityscapes** | | | | | | |
| Direct DINO | 68.42 | 66.81 | 87.18 | 0.11 | 96.93 | 2.94 |
| PCA (1152) | 68.10 | 67.28 | 85.73 | 0.12 | 96.98 | 2.91 |
| PCA (16) | 54.66 | 49.65 | 81.60 | 0.16 | 94.76 | 3.50 |
| VAE (16) | 65.64 | 64.31 | 84.75 | 0.12 | 96.94 | 2.92 |
| **Kubric** | | | | | | |
| Direct DINO | 99.54 | 99.16 | 77.63 | 0.16 | 99.62 | 0.29 |
| PCA (1152) | 99.49 | 99.08 | 77.52 | 0.15 | 99.62 | 0.29 |
| PCA (16) | 97.43 | 95.34 | 77.46 | 0.16 | 99.07 | 0.45 |
| VAE (16) | 99.17 | 98.49 | 78.02 | 0.16 | 99.56 | 0.30 |

These are however crucial for high-fidelity dense decoding, such as RGB (Figure 7).

To quantify the information loss, we train modality-specific decoders on autoencoded features from each method and evaluate their accuracy on ground truth data. Table 3 presents the results, indicating that PCA performs significantly worse than VAE on downstream dense prediction.

We further demonstrate the advantages of VAE in Section 4.3, where we show that our VAE improves *ReDi* (Kouzelis et al., 2025b), the state-of-the-art method for joint DINO PCA and RGB image generation.

It is worth noting that high-rank PCA preserves fine-grained pixel-level details but reintroduces geometry errors similar to those observed with direct diffusion. In other words, naively increasing latent capacity preserves information at the expense of generation quality (Yao et al., 2025), justifying the need for a more sophisticated autoencoder.

**Spectral properties of DINO features match RGB.**
Recently, the works of (Skorokhodov et al., 2025; Kouzelis et al., 2025a) show that modern RGB autoencoders over-represent high-frequency information in their latent spaces. This creates a mismatch with the *coarse-to-fine nature* (Dieleman, 2024; Wang et al., 2026; Falck et al., 2025) of denoising diffusion, harming sample quality. Bringing the frequency spectra of the latent closer to RGB substantially improves generation quality (Skorokhodov et al., 2025). Motivated by these results, we ask: *"Do these findings from the RGB domain transfer to DINO features?"*.

To this end, we first perform spectral decomposition of DINO features or their latents, quantifying the power of each DCT basis function sorted by its *zig-zag* frequency. Figure 6 shows that DINO features exhibit spectral power laws similar to RGB. Prior works (Karras et al., 2024; Chen, 2023) have noted the crucial importance of *signal-to-noise* ratio (SNR), heavily affected by latent scale. However, naively increasing KL regularization to constrain the latent scale shifts the spectrum away from RGB, reflecting the higher amount of noise in latents. However, unlike the PCA, VAE allows us to control the shift in spectrum by adjusting the strength of posterior regularization, balancing reconstruction ability, latent scale, and spectral properties. Concretely, we use $\beta = 0.01$ in all experiments.

### 4.3. Guiding Image Generation with VFMF VAE

We show the benefits of our VFM-VAE by applying it to *ReDi* image generator. Recall that *ReDi* uses PCA to compress DINO features as an additional target for generation, along with the RGB latents. They show, remarkably, that predicting RGB+DINO jointly improves the image generation quality, related to the findings of (Li et al., 2024).

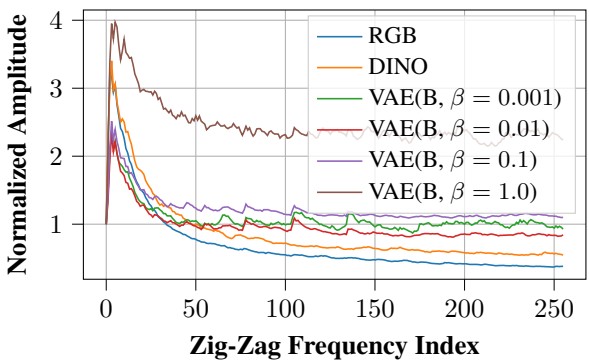

*Figure 6.* **Frequency profiles** on ImageNet 256x256. Uncompressed DINO features exhibit spectral characteristics similar to RGB inputs. As Gaussian regularization on the compressed features increases, the spectrum shifts toward higher frequencies, reflecting noise injected into the latent space.

However, while *ReDi* uses a standard latent space for RGB, they employ PCA to reduce the dimensionality of DINO. We instead train a VAE of the same dimensionality on *ImageNet* (Deng et al., 2009) and use the resulting VAE-compressed features as a drop-in replacement. For fairness, we train both variants from scratch with the SiT backbone (Ma et al., 2024) at two scales (SiT-B, SiT-XL), matching data, optimizer, and budget (400k updates), evaluating with the standard ADM (Karras et al., 2024) protocol. We evaluate quality of compressed features by applying *representation guidance* (Kouzelis et al., 2025b) during sampling.

**Results.** Table 4 reports consistent improvements for our VAE-guided variant over the PCA-guided baseline at 400k steps, across SiT-B and SiT-XL and for multiple values of the VAE KL weight $\beta$. Qualitative comparisons on SiT-XL (Figure 7) show sharper textures and better semantic faithfulness when guiding with VAE-compressed VFM features. Moreover, on SiT-B, our variant converges faster (Figure 8), indicating that our VAE-compressed DINO latents provide a stronger guidance signal than their PCA counterparts.

*Table 4.* **Image quality on conditional ImageNet 256×256.** Results are reported with optimal representation guidance scale $w_r$, after 400K training iterations. Diffusing our VAE latents instead of PCA projections results in higher-quality samples that better match the diversity of the ground truth distribution.

| Method | FID ($\downarrow$) | sFID ($\downarrow$) | Prec. ($\uparrow$) | Rec. ($\uparrow$) |
|---|---|---|---|---|
| SiT-B | | | | |
| *ReDi* (PCA) | 18.49 | 6.33 | 0.58 | 0.65 |
| *ReDi* (VAE, $\beta$=0.01) | **11.76** | 5.53 | 0.58 | **0.73** |
| *ReDi* (VAE, $\beta$=0.001) | 12.63 | **5.37** | **0.60** | 0.71 |
| SiT-XL | | | | |
| *ReDi* (PCA) | 5.48 | 4.66 | 0.59 | 0.77 |
| *ReDi* (VAE, $\beta$=0.01) | 5.01 | **4.48** | 0.61 | 0.77 |
| *ReDi* (VAE, $\beta$=0.001) | **4.98** | 4.55 | **0.61** | **0.77** |

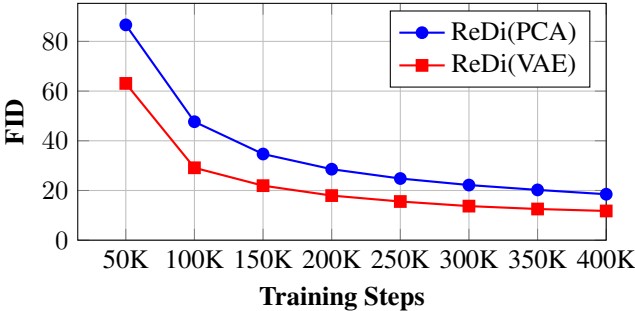

*Figure 8.* **FID on conditional ImageNet 256x256.** Replacing PCA ($c = 8$) projections with our VAE latents ($c = 8, \beta = 0.01$) in *ReDi* (Kouzelis et al., 2025b), applied to SiT-B (Ma et al., 2024), yields faster convergence and consistently better generation quality.

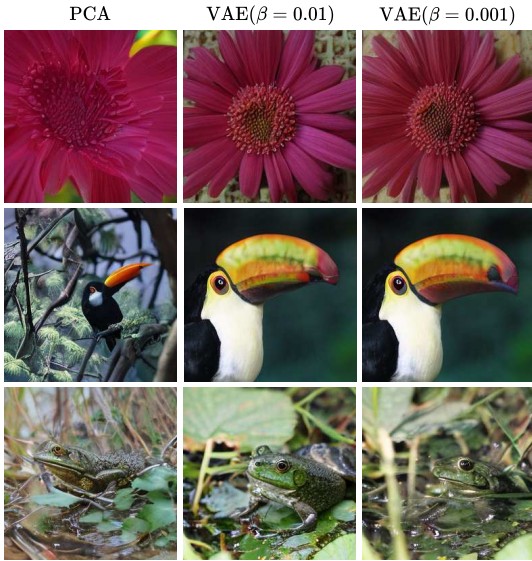

*Figure 7.* **Qualitative comparison of image quality** of **SiT-XL**, with representation guidance at 400K training steps. Diffusing VAE latents instead of PCA projections enhances fidelity, realism, and sharpness, resulting in higher quality samples.

## 5. Conclusion

We study scene forecasting from variable number of input frames and show that deterministic regression in VFM feature space averages over uncertain futures, degrading accuracy. We address this by generating autoregressively in a compact latent space of VFM features, yielding uncertainty-aware, sharper predictions from variable number of input frames. Our key insight is that generative modeling of VFM features requires (auto)encoding into a compact latent space suitable for diffusion. Across multiple modalities, our method outperforms regression and diffusion baselines, suggesting that stochastic generation of VFM features is a promising foundation for scalable scene forecasting, provided that these features are compressed optimally.

## Impact Statement

This paper presents work whose goal is to advance the field of machine learning, with a particular focus on generalized visual forecasting. The proposed approach has the potential to benefit research in areas such as robotics, simulation, and embodied visual understanding. While the method may have broader downstream applications, we do not identify any immediate societal impacts that require specific discussion here.

## Acknowledgements

We thank Jensen Zhou, Lorenza Prospero and Yiming Chen for useful discussions.

We acknowledge the use of resources provided by the Isambard-AI National AI Research Resource (AIRR). Isambard-AI (McIntosh-Smith et al., 2024) is operated by the University of Bristol and is funded by the UK Government's Department for Science, Innovation and Technology (DSIT) via UK Research and Innovation; and the Science and Technology Facilities Council [ST/AIRR/I-A-I/1023].

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

# Appendix

We also provide an **offline sample viewer website** in the `samples` folder of the supplementary archive.

## A. Ablations and Design Choices

For computational reasons, we conduct ablations only on *Cityscapes*, the smallest real dataset in our study.

**Choice of diffusion space.** Tables 5 and 6 show that diffusing VAE latents outperforms the alternatives.

**VAE**

*Table 7.* **Dense reconstruction accuracy** of different VAE architectures, model sizes, and regularization strength.

| Model | Segmentation | | Depth | | Normals | |
|---|---|---|---|---|---|---|
| | All ($\uparrow$) | Mov. ($\uparrow$) | d1 ($\uparrow$) | AbsRel ($\downarrow$) | a3 ($\uparrow$) | MeanAE ($\downarrow$) |
| Uncompressed DINO | 68.422 | 66.806 | 87.177 | 0.113 | 96.926 | 2.939 |
| PCA-compressed (rank=16) | 54.656 | 49.648 | 80.328 | 0.158 | 94.760 | 3.499 |
| VAE (ConvL, $\beta = 0.01$, d=16) | 65.642 | 64.310 | 84.754 | 0.123 | 96.942 | 2.918 |
| *Architecture & Model Size ($\beta = 0$)* | | | | | | |
| ConvNeXt-S | 61.022 | 59.025 | 84.848 | 0.145 | 96.845 | 2.976 |
| ConvNeXt-B | 64.303 | 62.484 | 86.189 | 0.129 | 97.007 | 2.910 |
| ConvNeXt-L | 65.337 | 63.571 | 86.606 | 0.122 | 97.102 | 2.885 |
| ViT-B | 65.597 | 63.840 | 86.991 | 0.123 | 97.048 | 2.903 |
| ViT-L | 65.910 | 64.168 | 87.170 | 0.117 | 97.080 | 2.890 |
| *KL Regularization (ConvNeXt-B)* | | | | | | |
| $\beta = 0.001$ | 64.090 | 62.257 | 86.451 | 0.131 | 97.070 | 2.901 |
| $\beta = 0.01$ | 63.959 | 62.116 | 85.901 | 0.130 | 96.991 | 2.916 |
| $\beta = 0.1$ | 63.960 | 62.116 | 85.901 | 0.130 | 96.991 | 2.916 |

**Architecture.** While ViT-based autoencoders demonstrate superior reconstruction accuracy, with ViT-B (65.60 mIoU) outperforming both the similarly sized ConvNeXt-B (64.30) and the larger ConvNeXt-L (65.34), they introduce significant practical overhead. Performance gains, likely driven by the global receptive field and self-attention mechanisms of the ViT architecture, are offset by the fact that they require more than double the GPU memory and exhibit slower training speeds. Given that these improvements are relatively marginal compared to markedly increased resource demands, we opt for convolutional autoencoders to maintain a better balance between reconstruction quality and computational efficiency.

**Model Size.** Scaling the network architecture shows a clear and consistent improvement of reconstruction quality. Within the ConvNeXt family, moving from the Small (S) to Large (L) variant results in a significant boost in semantic segmentation (from 61.02 to 65.34 mIoU) and a notable reduction in depth error (AbsRel decreasing from 0.145 to 0.122). A similar trend is observed in ViT-based autoencoders, where the ViT-L achieves the highest overall reconstruction quality, nearly matching the results of the uncompressed DINO baseline. This shows that increasing model capacity allows the VAE to better preserve high-frequency details and semantic information that are often lost during the bottleneck compression stage.

**KL Regularization.** As the regularization increases from $\beta = 0$ (unregularized) to $\beta = 0.1$ for the ConvNeXt-B model, there is a marginal degradation in performance. For instance, mIoU drops from 64.30 at $\beta = 0$ to approximately 63.96 at higher $\beta$ values. The performance appears to plateau between $\beta = 0.01$ and $\beta = 0.1$, suggesting that once a certain level of regularization is applied, further increases do not significantly impact the downstream task utility. However, further increase of $\beta$ shifts the latent spectrum (Figure 10) away from RGB, harming the *coarse-to-fine* nature of denoising diffusion. These results are consistent with recent findings in RGB domain (Skorokhodov et al., 2025). Thus, we opt for $\beta = 0.01$.

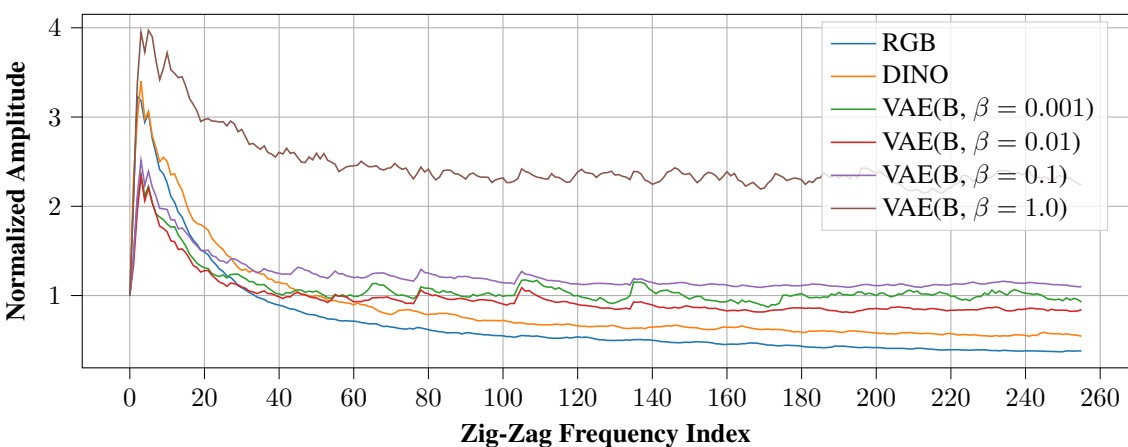

*Figure 9.* **Frequency profiles** on ImageNet 256x256. Uncompressed DINO features exhibit spectral characteristics similar to RGB inputs. As Gaussian regularization on the compressed features increases, the spectrum shifts toward higher frequencies, reflecting noise injected into the latent space.

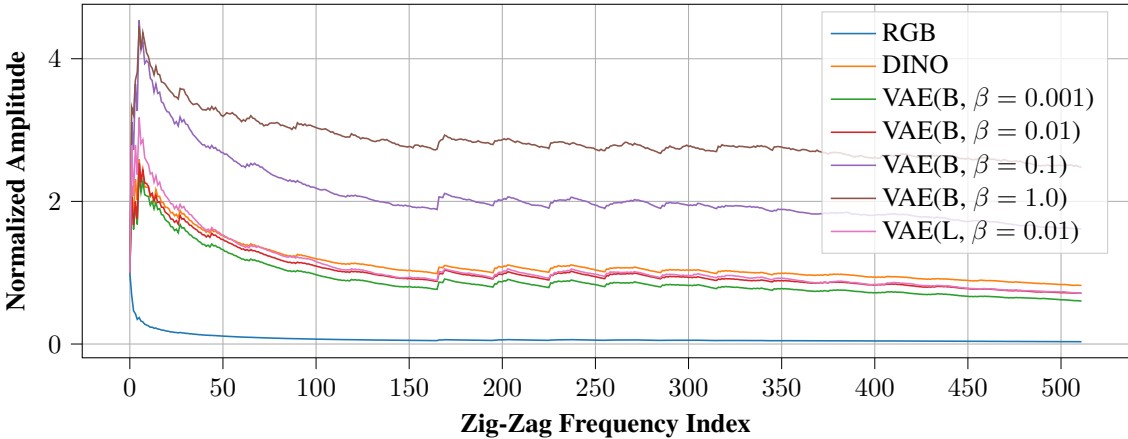

*Figure 10.* **Frequency profiles** on CityScapes. Uncompressed DINO features exhibit spectral characteristics similar to RGB inputs. As Gaussian regularization on the compressed features increases, the spectrum shifts toward higher frequencies, reflecting noise injected into the latent space.

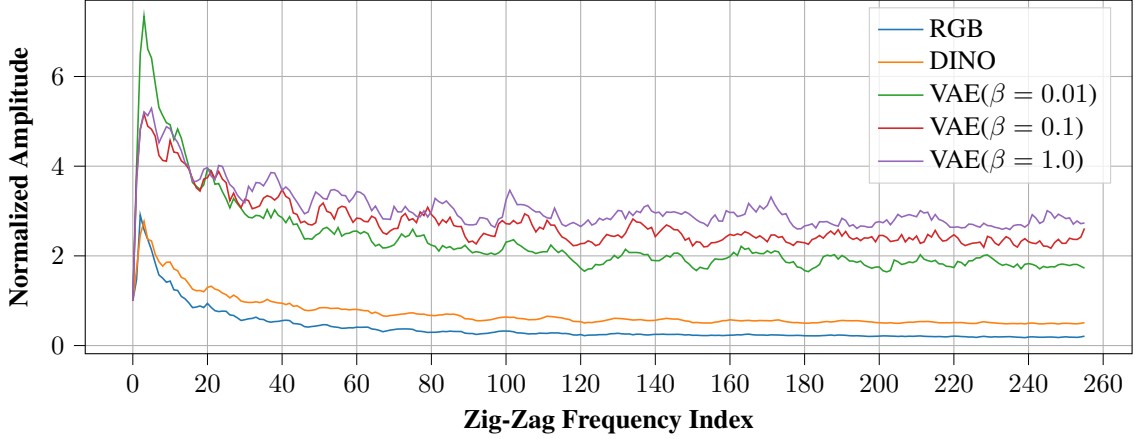

*Figure 11.* **Frequency profiles** on Kubric. Uncompressed DINO features exhibit spectral characteristics similar to RGB inputs. As Gaussian regularization on the compressed features increases, the spectrum shifts toward higher frequencies, reflecting noise injected into the latent space.

*Table 5.* **Dense forecasting accuracy** with different diffusion spaces on *CityScapes*. The VAE latent diffusion consistently delivers the best overall performance for dense forecasting.

| Model | Segmentation (mIoU) | | Depth | | Normals | |
|---|---|---|---|---|---|---|
| | All (↑) | Mov. (↑) | d1 (↑) | AbsRel (↓) | a3 (↑) | MeanAE (↓) |
| *initial context length $\|\mathcal{C}\|$ = 1, evaluation at 18th future frame after 9 autoregressive rollouts* | | | | | | |
| DINO-Foresight | 31.67 | 22.00 | 70.77 | 0.24 | 84.82 | 5.34 |
| **VAE (L, 16 channels)** | | | | | | |
| VFMF (Mean) | 34.85 | 31.76 | 75.89 | 0.19 | 88.48 | 4.55 |
| VFMF (Best) | 41.69 | 38.93 | 80.43 | 0.16 | 90.66 | 4.18 |
| **PCA (16 channels)** | | | | | | |
| VFMF (Mean) | 30.60 | 27.31 | 72.40 | 0.21 | 87.81 | 4.78 |
| VFMF (Best) | 30.16 | 26.92 | 77.61 | 0.19 | 89.04 | 4.57 |
| **PCA (1152 channels)** | | | | | | |
| VFMF (Mean) | 34.87 | 31.85 | 74.33 | 0.23 | 86.01 | 5.07 |
| VFMF (Best) | 33.79 | 30.73 | 76.43 | 0.21 | 86.94 | 4.91 |
| **Direct (3072 channels)** | | | | | | |
| VFMF (Mean) | 35.04 | 32.04 | 75.26 | 0.23 | 86.43 | 4.99 |
| VFMF (Best) | 33.72 | 30.67 | 77.11 | 0.21 | 87.54 | 4.85 |
| *initial context length $\|\mathcal{C}\|$ = 2, evaluation at 16th future frame after 8 autoregressive rollouts* | | | | | | |
| DINO-Foresight | 39.35 | 32.26 | 74.34 | 0.21 | 87.14 | 4.86 |
| **VAE (L, 16 channels)** | | | | | | |
| VFMF (Mean) | 41.74 | 38.96 | 78.63 | 0.17 | 90.87 | 4.09 |
| VFMF (Best) | 45.23 | 42.60 | 81.93 | 0.15 | 92.14 | 3.88 |
| **PCA (16 channels)** | | | | | | |
| VFMF (Mean) | 36.34 | 33.28 | 75.24 | 0.19 | 89.65 | 4.44 |
| VFMF (Best) | 35.72 | 32.67 | 78.95 | 0.18 | 90.51 | 4.29 |
| **PCA (1152 channels)** | | | | | | |
| VFMF (Mean) | 36.94 | 34.00 | 75.54 | 0.22 | 86.71 | 4.94 |
| VFMF (Best) | 36.24 | 33.28 | 77.48 | 0.20 | 87.52 | 4.79 |
| **Direct (3072 channels)** | | | | | | |
| VFMF (Mean) | 38.96 | 36.10 | 77.33 | 0.20 | 87.93 | 4.71 |
| VFMF (Best) | 37.47 | 34.55 | 79.07 | 0.19 | 88.75 | 4.62 |
| *initial context length $\|\mathcal{C}\|$ = 3, evaluation at 14th future frame after 7 autoregressive rollouts* | | | | | | |
| DINO-Foresight | 41.89 | 35.61 | 75.91 | 0.19 | 88.42 | 4.61 |
| **VAE (L, 16 channels)** | | | | | | |
| VFMF (Mean) | 44.53 | 41.87 | 79.72 | 0.16 | 91.78 | 3.92 |
| VFMF (Best) | 47.39 | 44.84 | 82.72 | 0.15 | 92.88 | 3.73 |
| **PCA (16 channels)** | | | | | | |
| VFMF (Mean) | 38.59 | 35.62 | 76.43 | 0.19 | 90.42 | 4.30 |
| VFMF (Best) | 37.73 | 34.75 | 79.81 | 0.17 | 91.21 | 4.16 |
| **PCA (1152 channels)** | | | | | | |
| VFMF (Mean) | 38.96 | 36.09 | 76.90 | 0.21 | 87.80 | 4.73 |
| VFMF (Best) | 38.13 | 35.23 | 78.80 | 0.19 | 88.54 | 4.60 |
| **Direct (3072 channels)** | | | | | | |
| VFMF (Mean) | 41.58 | 38.82 | 78.72 | 0.19 | 89.18 | 4.47 |
| VFMF (Best) | 40.18 | 37.36 | 80.60 | 0.18 | 89.92 | 4.39 |
| *initial context length $\|\mathcal{C}\|$ = 4, evaluation at 12th future frame after 6 autoregressive rollouts* | | | | | | |
| DINO-Foresight | 44.75 | 38.93 | 77.66 | 0.18 | 89.87 | 4.31 |
| **VAE (L, 16 channels)** | | | | | | |
| VFMF (Mean) | 46.49 | 43.90 | 80.72 | 0.16 | 92.53 | 3.77 |
| VFMF (Best) | 49.43 | 46.97 | 83.56 | 0.14 | 93.59 | 3.59 |
| **PCA (16 channels)** | | | | | | |
| VFMF (Mean) | 40.26 | 37.35 | 77.46 | 0.18 | 91.10 | 4.17 |
| VFMF (Best) | 39.45 | 36.53 | 80.58 | 0.16 | 91.92 | 4.02 |
| **PCA (1152 channels)** | | | | | | |
| VFMF (Mean) | 41.74 | 38.99 | 78.12 | 0.19 | 89.01 | 4.49 |
| VFMF (Best) | 40.97 | 38.19 | 79.90 | 0.18 | 89.67 | 4.38 |
| **Direct (3072 channels)** | | | | | | |
| VFMF (Mean) | 44.48 | 41.85 | 80.09 | 0.18 | 90.35 | 4.24 |
| VFMF (Best) | 42.80 | 40.10 | 81.89 | 0.17 | 90.98 | 4.18 |

*Table 6.* **Dense forecasting accuracy** with different diffusion spaces on *Kubric*. The VAE latent diffusion consistently delivers the best overall performance for dense forecasting.

| Model | Segmentation | | Depth | | Normals | |
|---|---|---|---|---|---|---|
| | All (↑) | Mov. (↑) | d1 (↑) | AbsRel (↓) | a3 (↑) | MeanAE (↓) |
| *initial context length $\|\mathcal{C}\| = 1$, evaluation at 22th future frame after 11 autoregressive rollouts* | | | | | | |
| DINO-Foresight | 46.73 | 5.11 | 64.37 | 0.24 | 90.62 | 2.94 |
| **VAE (B, 16 channels)** | | | | | | |
| VFMF (Mean) | 48.29 | 6.61 | 68.33 | 0.21 | 93.29 | 2.14 |
| VFMF (Best) | 70.55 | 47.77 | 88.80 | 0.08 | 93.45 | 2.07 |
| **PCA (16 channels)** | | | | | | |
| VFMF (Mean) | 49.68 | 14.48 | 68.33 | 0.28 | 88.91 | 3.30 |
| VFMF (Best) | 70.39 | 47.58 | 88.16 | 0.08 | 93.31 | 2.07 |
| **PCA (1152 channels)** | | | | | | |
| VFMF (Mean) | 48.76 | 9.30 | 65.95 | 0.22 | 90.96 | 2.75 |
| VFMF (Best) | 69.49 | 45.30 | 87.38 | 0.09 | 93.82 | 1.94 |
| **Direct (3072 channels)** | | | | | | |
| VFMF (Mean) | 48.48 | 10.21 | 68.49 | 0.22 | 90.73 | 2.86 |
| VFMF (Best) | 69.45 | 46.25 | 87.79 | 0.09 | 93.12 | 2.18 |
| *initial context length $\|\mathcal{C}\| = 2$, evaluation at 20th future frame after 10 autoregressive rollouts* | | | | | | |
| DINO-Foresight | 51.15 | 14.40 | 62.60 | 0.24 | 89.20 | 3.36 |
| **VAE (B, 16 channels)** | | | | | | |
| VFMF (Mean) | 55.89 | 21.68 | 68.87 | 0.23 | 92.31 | 2.39 |
| VFMF (Best) | 64.84 | 37.97 | 78.06 | 0.16 | 91.66 | 2.59 |
| **PCA (16 channels)** | | | | | | |
| VFMF (Mean) | 55.50 | 22.62 | 66.12 | 0.29 | 89.51 | 3.12 |
| VFMF (Best) | 62.65 | 34.40 | 77.72 | 0.17 | 91.15 | 2.74 |
| **PCA (1152 channels)** | | | | | | |
| VFMF (Mean) | 52.02 | 14.51 | 62.26 | 0.24 | 91.17 | 2.68 |
| VFMF (Best) | 58.84 | 26.88 | 71.43 | 0.20 | 91.51 | 2.65 |
| **Direct (3072 channels)** | | | | | | |
| VFMF (Mean) | 54.54 | 19.61 | 63.13 | 0.24 | 91.09 | 2.71 |
| VFMF (Best) | 60.70 | 30.48 | 78.04 | 0.17 | 91.26 | 2.71 |
| *initial context length $\|\mathcal{C}\| = 3$, evaluation at 18th future frame after 9 autoregressive rollouts* | | | | | | |
| DINO-Foresight | 54.28 | 19.96 | 66.76 | 0.22 | 89.43 | 3.26 |
| **VAE (B, 16 channels)** | | | | | | |
| VFMF (Mean) | 59.47 | 27.79 | 71.94 | 0.21 | 92.67 | 2.26 |
| VFMF (Best) | 68.25 | 43.74 | 80.09 | 0.14 | 92.31 | 2.40 |
| **PCA (16 channels)** | | | | | | |
| VFMF (Mean) | 58.69 | 28.07 | 68.51 | 0.26 | 89.99 | 2.97 |
| VFMF (Best) | 65.94 | 39.97 | 78.73 | 0.15 | 91.80 | 2.54 |
| **PCA (1152 channels)** | | | | | | |
| VFMF (Mean) | 53.96 | 18.14 | 64.74 | 0.23 | 91.20 | 2.68 |
| VFMF (Best) | 60.59 | 30.08 | 71.73 | 0.19 | 91.62 | 2.61 |
| **Direct (3072 channels)** | | | | | | |
| VFMF (Mean) | 56.35 | 22.98 | 66.11 | 0.23 | 91.12 | 2.70 |
| VFMF (Best) | 62.48 | 33.87 | 79.52 | 0.15 | 91.46 | 2.67 |
| *initial context length $\|\mathcal{C}\| = 4$, evaluation at 16th future frame after 8 autoregressive rollouts* | | | | | | |
| DINO-Foresight | 57.62 | 25.78 | 69.31 | 0.22 | 89.82 | 3.11 |
| **VAE (B, 16 channels)** | | | | | | |
| VFMF (Mean) | 61.74 | 31.86 | 71.88 | 0.21 | 92.66 | 2.25 |
| VFMF (Best) | 69.86 | 46.56 | 80.50 | 0.14 | 92.62 | 2.31 |
| **PCA (16 channels)** | | | | | | |
| VFMF (Mean) | 60.66 | 31.27 | 69.26 | 0.25 | 90.57 | 2.81 |
| VFMF (Best) | 68.04 | 43.52 | 79.44 | 0.15 | 92.19 | 2.42 |
| **PCA (1152 channels)** | | | | | | |
| VFMF (Mean) | 54.82 | 19.86 | 67.19 | 0.23 | 91.22 | 2.69 |
| VFMF (Best) | 62.05 | 32.94 | 74.39 | 0.17 | 91.61 | 2.63 |
| **Direct (3072 channels)** | | | | | | |
| VFMF (Mean) | 57.86 | 25.81 | 67.93 | 0.22 | 91.01 | 2.72 |
| VFMF (Best) | 64.39 | 37.25 | 80.79 | 0.15 | 91.53 | 2.64 |

**Number of samples ($k$).** Tables 8 to 11 demonstrate that our generative method outperforms or matches *DINO-Foresight* on majority of metrics, even when using a single sample ($k = 1$). Furthermore, performance gains from generative approach increase substantially as the number of samples ($k$) grows. These findings confirm the superiority of our generative formulation, and show that in practice, only a modest number of samples ($k \in \{2, \ldots, 8\}$) is sufficient to achieve considerably more accurate forecasts than the baseline. These conclusions hold for all context lengths, $|\mathcal{C}| \in \{1, 2, 3, 4\}$.

*Table 8.* **Dense Forecasting Accuracy** for different number of samples ($k$) on *Cityscapes* for context length $|\mathcal{C}| = 1$.

| Model | Segmentation | | Depth | | Normals | |
|---|---|---|---|---|---|---|
| | All (↑) | Mov. (↑) | d1 (↑) | AbsRel (↓) | a3 (↑) | MeanAE (↓) |
| DINO-Foresight | 31.67 | 22.00 | 70.77 | 0.23 | 84.82 | 5.33 |
| DINO-Foresight (V) | 30.62 | 20.04 | 71.65 | 0.23 | 84.96 | 5.28 |
| **VFMF (Mean)** | | | | | | |
| $k = 1$ | 32.22 | 29.06 | 73.59 | 0.22 | 85.95 | 5.09 |
| $k = 2$ | 33.33 | 30.20 | 74.31 | 0.21 | 87.13 | 4.83 |
| $k = 4$ | 34.26 | 31.17 | 75.11 | 0.20 | 87.79 | 4.69 |
| $k = 8$ | 34.77 | 31.69 | 75.49 | 0.20 | 88.21 | 4.61 |
| $k = 16$ | 34.73 | 31.65 | 75.75 | 0.19 | 88.42 | 4.56 |
| $k = 32$ | 34.85 | 31.76 | 75.89 | 0.19 | 88.48 | 4.55 |
| **VFMF (Best)** | | | | | | |
| $k = 1$ | 32.22 | 29.06 | 73.59 | 0.22 | 85.95 | 5.09 |
| $k = 2$ | 34.65 | 31.60 | 75.45 | 0.20 | 87.43 | 4.79 |
| $k = 4$ | 36.17 | 33.17 | 77.25 | 0.19 | 88.52 | 4.58 |
| $k = 8$ | 37.82 | 34.88 | 78.52 | 0.18 | 89.43 | 4.41 |
| $k = 16$ | 39.61 | 36.75 | 79.65 | 0.17 | 90.16 | 4.27 |
| $k = 32$ | 41.70 | 38.93 | 80.43 | 0.16 | 90.66 | 4.18 |

*Table 9.* **Dense Forecasting Accuracy** for different number of samples ($k$) on *Cityscapes* for context length $|\mathcal{C}| = 2$.

| Model | Segmentation | | Depth | | Normals | |
|---|---|---|---|---|---|---|
| | All (↑) | Mov. (↑) | d1 (↑) | AbsRel (↓) | a3 (↑) | MeanAE (↓) |
| DINO-Foresight | 39.35 | 32.26 | 74.34 | 0.21 | 87.14 | 4.86 |
| DINO-Foresight (V) | 40.13 | 31.92 | 77.14 | 0.19 | 88.90 | 4.50 |
| **VFMF (Mean)** | | | | | | |
| $k = 1$ | 39.03 | 36.12 | 77.48 | 0.19 | 89.84 | 4.32 |
| $k = 2$ | 39.94 | 37.07 | 78.02 | 0.18 | 90.37 | 4.21 |
| $k = 4$ | 41.10 | 38.29 | 78.34 | 0.18 | 90.60 | 4.16 |
| $k = 8$ | 41.47 | 38.67 | 78.47 | 0.17 | 90.73 | 4.12 |
| $k = 16$ | 41.68 | 38.89 | 78.53 | 0.17 | 90.83 | 4.10 |
| $k = 32$ | 41.74 | 38.96 | 78.63 | 0.17 | 90.87 | 4.09 |
| **VFMF (Best)** | | | | | | |
| $k = 1$ | 39.03 | 36.12 | 77.48 | 0.19 | 89.84 | 4.32 |
| $k = 2$ | 40.56 | 37.71 | 78.79 | 0.18 | 90.55 | 4.18 |
| $k = 4$ | 42.52 | 39.77 | 79.92 | 0.17 | 91.03 | 4.08 |
| $k = 8$ | 43.85 | 41.17 | 80.69 | 0.16 | 91.36 | 4.01 |
| $k = 16$ | 44.72 | 42.07 | 81.32 | 0.16 | 91.76 | 3.94 |
| $k = 32$ | 45.23 | 42.60 | 81.93 | 0.15 | 92.02 | 3.88 |

*Table 10.* **Dense Forecasting Accuracy** for different number of samples ($k$) on *Cityscapes* for context length $|\mathcal{C}| = 3$.

| Model | Segmentation | | Depth | | Normals | |
|---|---|---|---|---|---|---|
| | All (↑) | Mov. (↑) | d1 (↑) | AbsRel (↓) | a3 (↑) | MeanAE (↓) |
| DINO-Foresight | 41.89 | 35.61 | 75.91 | 0.19 | 88.42 | 4.60 |
| DINO-Foresight (V) | 44.48 | 38.21 | 79.14 | 0.17 | 90.71 | 4.16 |
| **VFMF (Mean)** | | | | | | |
| $k = 1$ | 42.33 | 39.57 | 78.81 | 0.18 | 90.79 | 4.14 |
| $k = 2$ | 43.09 | 40.36 | 79.36 | 0.17 | 91.29 | 4.04 |
| $k = 4$ | 43.77 | 41.07 | 79.58 | 0.17 | 91.52 | 3.98 |
| $k = 8$ | 44.26 | 41.58 | 79.61 | 0.17 | 91.67 | 3.95 |
| $k = 16$ | 44.52 | 41.86 | 79.70 | 0.17 | 91.73 | 3.93 |
| $k = 32$ | 44.53 | 41.87 | 79.72 | 0.16 | 91.78 | 3.92 |
| **VFMF (Best)** | | | | | | |
| $k = 1$ | 42.33 | 39.57 | 78.81 | 0.18 | 90.79 | 4.14 |
| $k = 2$ | 43.37 | 40.64 | 80.09 | 0.17 | 91.46 | 4.01 |
| $k = 4$ | 44.82 | 42.15 | 80.93 | 0.16 | 91.93 | 3.92 |
| $k = 8$ | 45.97 | 43.36 | 81.64 | 0.16 | 92.31 | 3.85 |
| $k = 16$ | 46.63 | 44.05 | 82.27 | 0.15 | 92.62 | 3.78 |
| $k = 32$ | 47.39 | 44.84 | 82.72 | 0.15 | 92.88 | 3.73 |

*Table 11.* **Dense Forecasting Accuracy** for different number of samples ($k$) on *Cityscapes* for context length $|\mathcal{C}| = 4$.

| Model | Segmentation | | Depth | | Normals | |
|---|---|---|---|---|---|---|
| | All (↑) | Mov. (↑) | d1 (↑) | AbsRel (↓) | a3 (↑) | MeanAE (↓) |
| DINO-Foresight | 44.75 | 38.92 | 77.66 | 0.18 | 89.87 | 4.31 |
| DINO-Foresight (V) | 47.17 | 40.94 | 80.51 | 0.16 | 92.05 | 3.89 |
| **VFMF (Mean)** | | | | | | |
| $k = 1$ | 44.20 | 41.51 | 80.00 | 0.17 | 91.69 | 3.97 |
| $k = 2$ | 45.02 | 42.37 | 80.24 | 0.16 | 92.13 | 3.87 |
| $k = 4$ | 45.86 | 43.25 | 80.50 | 0.16 | 92.31 | 3.82 |
| $k = 8$ | 46.24 | 43.65 | 80.53 | 0.16 | 92.42 | 3.80 |
| $k = 16$ | 46.48 | 43.89 | 80.67 | 0.16 | 92.51 | 3.77 |
| $k = 32$ | 46.49 | 43.90 | 80.72 | 0.16 | 92.53 | 3.77 |
| **VFMF (Best)** | | | | | | |
| $k = 1$ | 44.20 | 41.51 | 80.00 | 0.17 | 91.69 | 3.97 |
| $k = 2$ | 45.26 | 42.61 | 80.89 | 0.16 | 92.28 | 3.85 |
| $k = 4$ | 46.50 | 43.90 | 81.75 | 0.15 | 92.68 | 3.77 |
| $k = 8$ | 47.68 | 45.14 | 82.40 | 0.15 | 93.02 | 3.70 |
| $k = 16$ | 48.70 | 46.21 | 83.03 | 0.15 | 93.35 | 3.64 |
| $k = 32$ | 49.43 | 46.97 | 83.56 | 0.14 | 93.59 | 3.59 |

# B. Qualitative Results

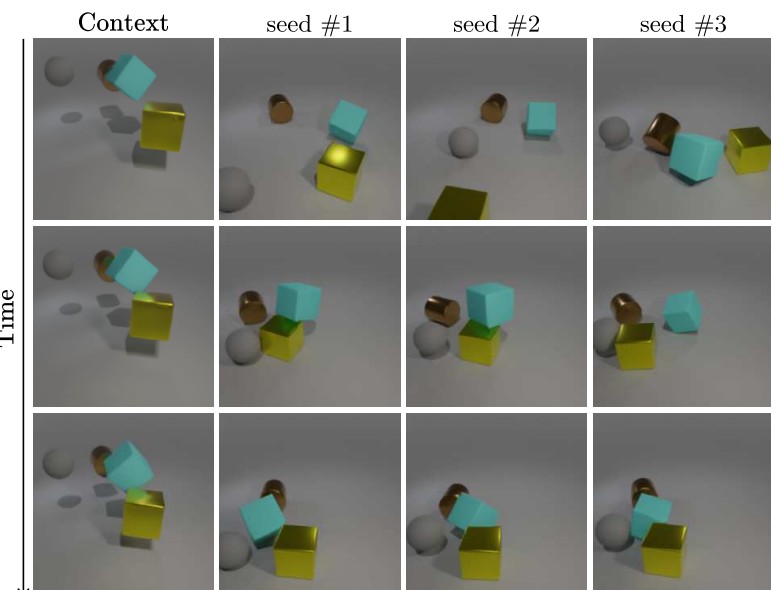

*(a)* **Future predictions on Kubric** decoded to RGB.

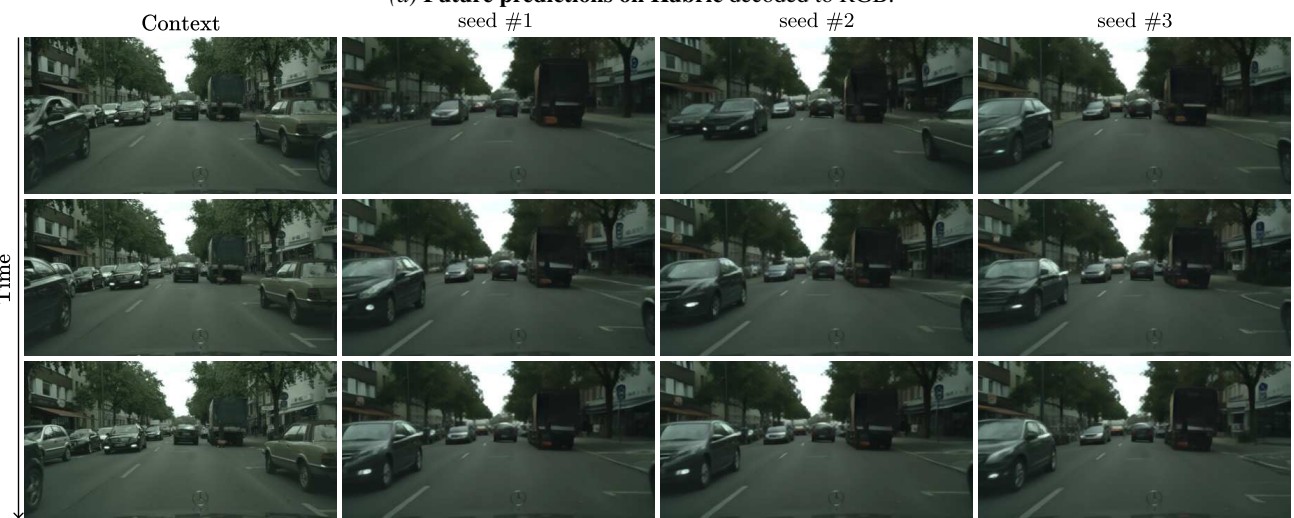

*(b)* **Future predictions on Cityscapes** decoded to RGB.

*Figure 12.* **Diversity of future predictions depends on context length.** As the number of input frames increases, diversity in future predictions decreases. With two frames, velocity can be inferred; with three frames, acceleration becomes observable. This pattern holds for both synthetic data (*Kubric*) and real data (*Cityscapes*). Predictions from shorter contexts are inherently more ambiguous: on *Kubric*, object velocities and accelerations are unknown; on real data, the velocities and accelerations of the ego vehicle and other vehicles are unknown. As more frames are observed, prediction diversity decreases, reflecting reduced uncertainty. This demonstrates that our model dynamically reduces uncertainty as more environmental information becomes available. Prior regression methods such as *DINO-Foresight*, *DINO-World*, and *DINO-WM* fundamentally lack this capability.

**Feature Forecasting.** Please refer to our **offline website** for more sample animations.

**Image generation.** Figures 13 to 16 demonstrate that diffusing VAE latents instead of PCA projections enhances fidelity, realism, and sharpness, yielding overall higher quality samples.

PCA          VAE($\beta = 0.01$)          VAE($\beta = 0.001$)

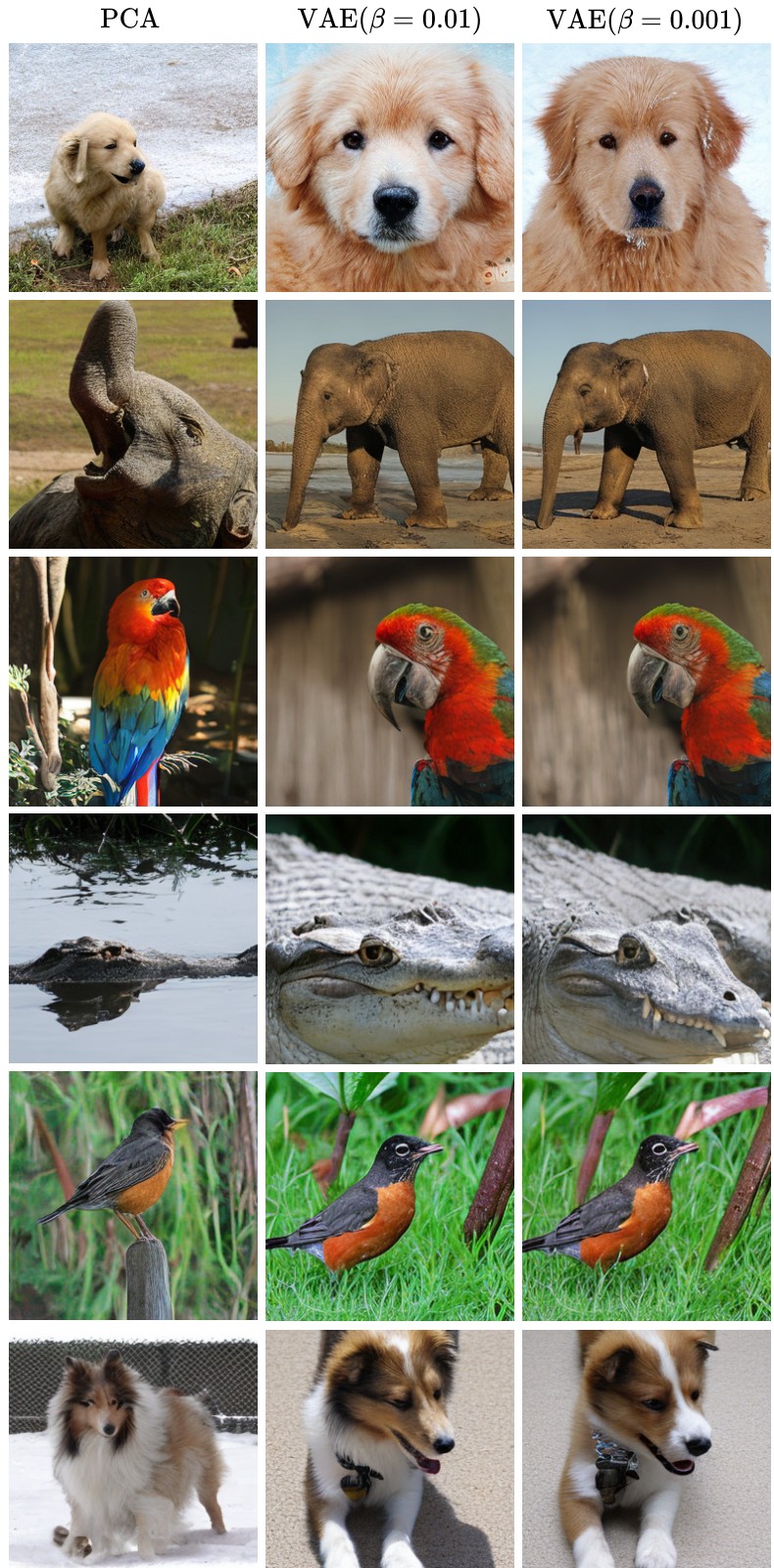

*Figure 13.* **Qualitative comparison of image quality** of **SiT-XL**, with ReDi guidance at 400K training steps. Diffusing VAE latents instead of PCA projections enhances fidelity, realism, and sharpness, resulting in higher quality samples.

PCA       VAE($\beta = 0.01$)       VAE($\beta = 0.001$)

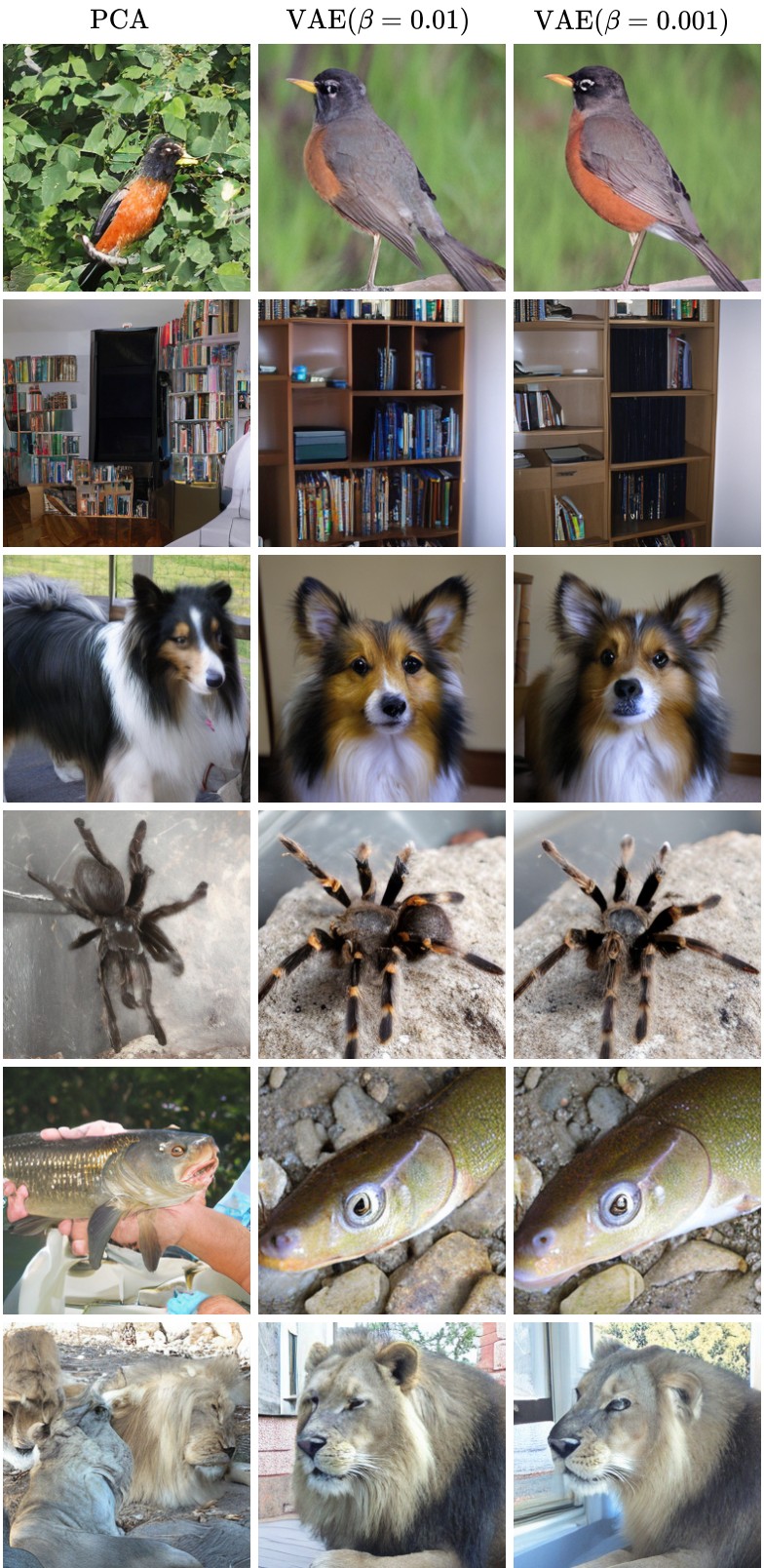

*Figure 14.* **Qualitative comparison of image quality** of **SiT-XL**, with ReDi guidance at 400K training steps. Diffusing VAE latents instead of PCA projections enhances fidelity, realism, and sharpness, resulting in higher quality samples.

PCA          VAE($\beta = 0.01$)          VAE($\beta = 0.001$)

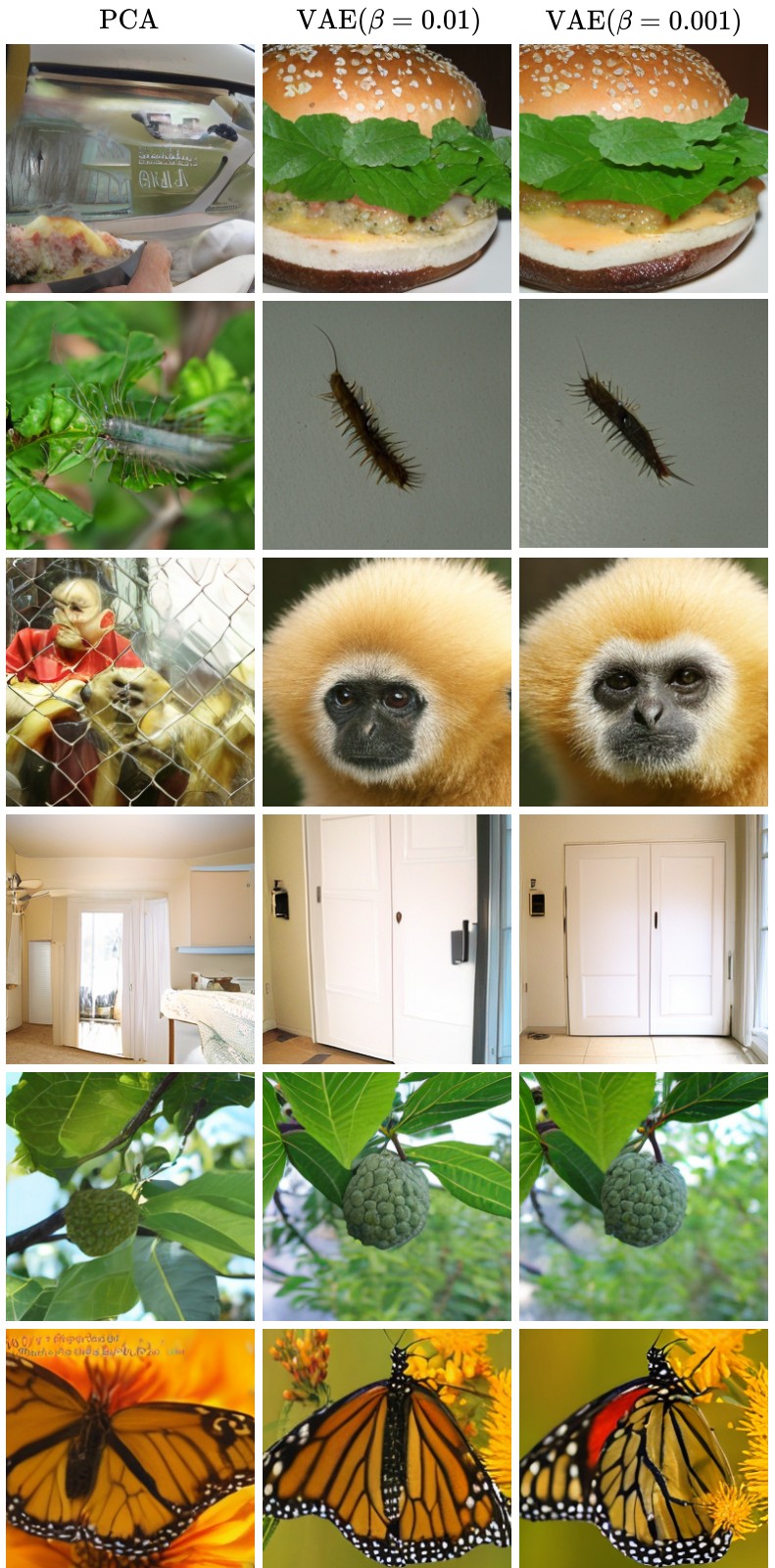

*Figure 15.* **Qualitative comparison of image quality** of **SiT-XL**, with ReDi guidance at 400K training steps. Diffusing VAE latents instead of PCA projections enhances fidelity, realism, and sharpness, resulting in higher quality samples.

PCA         VAE($\beta = 0.01$)         VAE($\beta = 0.001$)

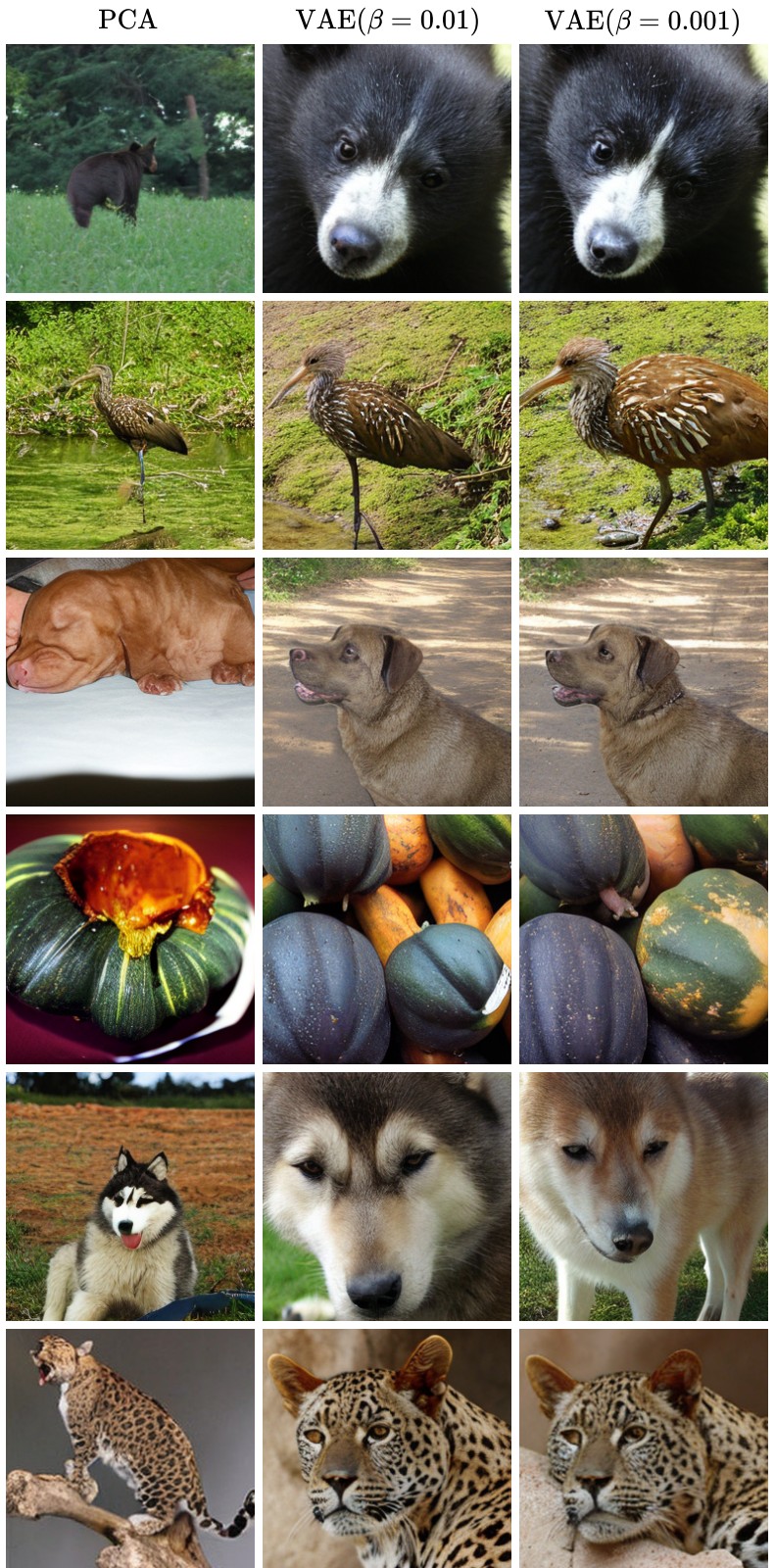

*Figure 16.* **Qualitative comparison of image quality** of **SiT-XL**, with ReDi guidance at 400K training steps. Diffusing VAE latents instead of PCA projections enhances fidelity, realism, and sharpness, resulting in higher quality samples.

# C. Implementation Details

## C.1. Multi-Scale Feature Space

It is well known that different layers of foundation models capture information at different scales (Bolya et al., 2025; Carreira et al., 2025; Amir et al., 2022). In particular, Bolya et al. demonstrates that the most informative visual features in vision foundation models (VFMs) often reside in intermediate layers. Motivated by this observation, we also work with features from multiple layers of VFM.

Specifically, we use the DINOv2 (Oquab et al., 2024) ViT-B(*with registers (Darcet et al., 2024)*), extracting and concatenating features from layers $3, 6, 9, 12$, following *DINO-Foresight* (Karypidis et al., 2026). While the resulting multi-scale feature space increases expressivity, it also substantially raises feature dimensionality. For instance, concatenated ViT-B features are 3072 dimensional. The concatenated features are further compressed by our proposed VAE to mitigate the curse of dimensionality.

## C.2. Feature VAE

Our feature VAE employs an encoder-decoder architecture with shared design principles, performing compression exclusively along the channel dimension. We explore both convolutional and transformer-based architectures in three sizes: S*mall*, B*ase*, and L*arge*. The convolutional VAEs build on the *isotropic* ConvNeXt architecture (Liu et al., 2022), while the transformer variants are based on the feature transformer from *DINO-Foresight* (Karypidis et al., 2026). Details about these architectures are in Tables 12 and 13. For compute constraints, we use convolutional VAEs since they require significantly less GPU memory to train.

The encoder operates as follows: it takes an input feature map $f \in \mathbb{R}^{H \times W \times D}$ and projects it linearly to the model dimension $f \in \mathbb{R}^{H \times W \times D_{\text{model}}}$. This representation is then processed through a sequence of isotropic layers (either transformer or convolutional blocks). Finally, a linear projection maps from $\mathbb{R}^{H \times W \times D_{\text{model}}}$ to $z \in \mathbb{R}^{H \times W \times 2 \times D_{\text{latent}}}$, producing the mean and log variance of the latent distribution.

The decoder mirrors the encoder architecture. It begins by projecting the latent representation $z \in \mathbb{R}^{H \times W \times \times D_{\text{latent}}}$ back to the model dimension $h \in \mathbb{R}^{H \times W \times D_{\text{model}}}$ via a linear projection. This representation is then processed through a sequence of isotropic layers (either transformer or convolutional blocks). Finally, a linear projection maps from $\mathbb{R}^{H \times W \times D_{\text{model}}}$ to $\hat{f} \in \mathbb{R}^{H \times W \times D}$.

The number of isotropic layers is the same in the encoder and decoder. For a fair comparison with *ReDi* (Kouzelis et al., 2025b), we set our VAE's latent dimensionality to match *ReDi*'s PCA rank (8 channels). Here, we also used only the final layer features same as *ReDi*. For a fair comparison with *DINO-Foresight*, we used a 16-channel VAE and considered PCA baselines with a rank of 16 or the official *DINO-Foresight* setting of a rank of 1152.

**Training.** We train all models using the `AdamW` optimizer (Kingma & Ba, 2015; Loshchilov & Hutter, 2019) with a learning rate of $3 \times 10^{-4}$ and linear warmup, gradient clipping at norm 1.0, and mixed precision (`bfloat16`). For Cityscapes and Kubric, we use an effective batch size of 256 across 4 or 8 L40s 40GB GPUs, training for 200 epochs on Cityscapes and 2000 epochs on Kubric (approximately 2 days each). For ImageNet, we train for 100 epochs. We use an effective batch size of 2048 distributed across 4 NVIDIA GH200 Grace Hopper Superchips. This takes approximately 1 day.

## C.3. Feature Denoiser

We deliberately make minimal changes to *DINO-Foresight*, a deterministic baseline. In particular, our denoising network architecture is almost the same as masked feature transformer from *DINO-Foresight*. The network uses 12 transformer layers with a hidden size of 1152. Each input sequence contains up to 4 context frames ($|\mathcal{C}| = 4$) and 1 noisy prediction frame, where the network denoises only the prediction frame.

We extend the original *DINO-Foresight* architecture with only two key components for denoising: (1) timestep encoding and (2) timestep injection.

**Timestep Encoding.** We introduce a flow matching time input $t \in [0, 1]$, which is processed through standard sinusoidal positional encoding with 256 frequencies. The encoded timestep is then projected to the transformer dimension via a 2-layer MLP.

**Timestep Injection.** Following DiT (Peebles & Xie, 2023) and SiT (Ma et al., 2024), we condition the network on the timestep embedding using zero-adaptive normalization (adaLN-Zero). Specifically, the timestep embedding passes through another 2-layer MLP that regresses 9 adaptive normalization parameters: shift, scale, and gate parameters for spatial attention, temporal attention, and the MLP. These parameters are shared across all transformer blocks, as in (Chen et al., 2023; Gao et al., 2024a; Lan et al., 2025).

Since *DINO-Foresight* employs spatial-temporal attention, we replicate the regressed normalization parameters along both spatial and temporal dimensions. The adaptive normalization is then applied after each spatial attention, temporal attention, and MLP operation within every transformer block. Finally, as in DiT and SiT, we add a final adaptive normalization layer after all transformer blocks, which regresses its own set of parameters.

**Training.** We use `AdamW` optimizer (Kingma & Ba, 2015; Loshchilov & Hutter, 2019) with a learning rate of $6.4 \times 10^{-4}$ with cosine annealing. Training is performed on 8 L40s 40GB GPUs, giving an effective batch size of 64. On Cityscapes, we train for 3200 epochs. On Kubric, we train for 2400 epochs. When training with variable context length, especially on Kubric, we encountered training instability with both *DINO-Foresight* and our modified architecture. The problem was due to exploding attention logits that we resolved with `QKNorm` (Henry et al., 2020). For fairness, we apply `QKNorm` to both *DINO-Foresight* and our method.

**Sampling**. We use the Euler solver with 10 NFEs for all the experiments.

### C.4. Probing heads

We adhere closely to the *DINO-Foresight* protocol. While we utilize the official training configuration for CityScapes (base learning rate of $10^{-4}$), we have observed training instability on Kubric. To address this, we decreased the base learning rate on Kubric to $10^{-5}$ for cosine annealing while keeping all other parameters unchanged.

We detail the architectures and training setups used for the various downstream tasks. For semantic segmentation, depth estimation, and surface normal prediction, we employ the DPT head (Ranftl et al., 2021). We set the feature dimension to 256 and configure `dpt_out_channels` = [128, 256, 512, 512].

All models are trained for 100 epochs with an effective batch size of 128 distributed across either 4 or 8 GPUs. Optimization is performed using AdamW with a learning rate of $1.6 \times 10^{-3}$, linear warmup during the first 10 epochs, and a weight decay of $10^{-4}$. We tailor the loss functions and schedulers to each task.
*Semantic Segmentation:* We apply a polynomial learning-rate scheduler and optimize with cross-entropy over 19 classes (CityScapes) or 2 classes (Kubric).
*Depth Estimation:* We use a cosine annealing scheduler and cross-entropy loss with 256 classes.
*Surface Normal Prediction:* We use a polynomial scheduler and a loss combining cosine similarity and $L_2$ distance with weighted averaging over 3 classes.

*Table 12.* **Hyperparameters for ViT (Transformer) VAE Architecture**.

| Hyperparameter | **Base** (B) | **Large** (L) |
|---|---|---|
| dinov2_variant | "vitb14_reg" | "vitb14_reg" |
| intermediate_layers | [2, 5, 8, 11] | [2, 5, 8, 11] |
| patch_size | 14 | 14 |
| input_dim | 3072 | 3072 |
| latent_channels | 16 | 16 |
| dropout | 0.1 | 0.1 |
| use_qk_norm | true | true |
| abs_pos_enc | true | true |
| num_encoder_layers | 12 | 24 |
| num_decoder_layers | 12 | 24 |
| heads | 12 | 16 |
| hidden_dim | 768 | 1024 |
| mlp_dim | 3072 | 4096 |
| num_registers | 4 | 4 |

*Table 13.* **Hyperparameters for ConvNeXt Isotropic VAE Architecture**.

| Hyperparameter | Small (S) | Base(B) | Large (L) |
|---|---|---|---|
| dinov2_variant | "vitb14_reg" | "vitb14_reg" | "vitb14_reg" |
| intermediate_layers | [2,5,8,11] | [2,5,8,11] | [2,5,8,11] |
| patch_size | 14 | 14 | 14 |
| input_dim | 3072 | 3072 | 3072 |
| latent_channels | 16 | 16 | 16 |
| drop_path_rate | 0 | 0 | 0 |
| layer_scale_init_value | 0 | 0 | 0 |
| depth | 18 | 18 | 36 |
| dim | 384 | 768 | 1024 |

### C.5. RGB Decoder

The RGB-Decoder uses a transformer backbone followed by a DPT-Head (Ranftl et al., 2021) with default parameters. The transformer backbone is identical to the *DINO-Foresight* ViT encoder (Table 12), using the *base* configuration for Kubric and CityScapes experiments and the *large* configuration for ImageNet. Features are extracted from transformer backbone layers [2, 5, 8, 11] and processed by the DPT head using the VGG-T (Wang et al., 2025) implementation, resulting in 3 RGB channels. The decoder is trained with an equally-weighted combination of L1 and LPIPS.

## D. Evaluation

### D.1. Datasets

**Cityscapes.**
*Cityscapes* (Cordts et al., 2016) provides 2,975 training and 500 validation sequences, each 30 frames at 16 fps with resolution $1024 \times 2048$; following *DINO-Foresight*, we downsample them to $224 \times 448$ for computational efficiency. Only the $20^{th}$ frame comes with dense semantic labels for 19 classes.

**Kubric.**
*Kubric MOVi-A* (Greff et al., 2022) contains 9,703 training and 250 validation sequences, each of 24 frames at 12 fps and $256 \times 256$ resolution, depicting 3–10 rigid objects moving on a static background with collisions; full per-frame annotations (segmentation, depth, flow, 3D) are available.

**ScanNet.**
*ScanNet* (Dai et al., 2017) is an RGB-D video dataset consisting of about 2 million views in more than 1500 scenes, annotated with 3D camera poses, surface reconstructions, depth and instance-level semantic segmentation. Scenes consist of variable number of frames, often several hundreds, recorded at 30 FPS, with RGB resolution of $1296 \times 968$ and depth of $640 \times 480$. Training set consists of 1201 scenes, validation of 312 and test of 100 scenes.

### D.2. Protocol

**Cityscapes.**
We evaluate our predictions on the official validation split.

- Given input frames: $[1]$, forecast frame 19 by autoregressively rolling out $[3, 5, 7, 9, 11, 13, 15, 17, 19]$.
- Given input frames: $[1, 3]$, forecast frame 19 by autoregressively rolling out $[5, 7, 9, 11, 13, 15, 17, 19]$.
- Given input frames: $[1, 3, 5]$, forecast frame 19 by autoregressively rolling out $[7, 9, 11, 13, 15, 17, 19]$.
- Given input frames: $[1, 3, 5, 7]$, forecast frame 19 by autoregressively rolling out $[9, 11, 13, 15, 17, 19]$.

**Kubric.**
For evaluation, we additionally construct an unseen test set of 128 scenes and generate 64 distinct futures per scene by varying initial object velocities while holding initial poses fixed.

- Given input frames: $[0]$, forecast frame 22 by autoregressively rolling out $[2, 4, 6, 8, 10, 12, 14, 16, 18, 20, 22]$.

- Given input frames: $[0, 2]$, forecast frame 22 by autoregressively rolling out $[4, 6, 8, 10, 12, 14, 16, 18, 20, 22]$.
- Given input frames: $[0, 2, 4]$, forecast frame 22 by autoregressively rolling out $[6, 8, 10, 12, 14, 16, 18, 20, 22]$.
- Given input frames: $[0, 2, 4, 6]$, forecast frame 22 by autoregressively rolling out $[8, 10, 12, 14, 16, 18, 20, 22]$.

**ScanNet.**
We evaluate our predictions on the official validation split.

- Given input frames: $[0, 1, 2, 3]$, forecast frame 15 by autoregressively rolling out $[4, 5, 6, 7, 8, 9, 10, 11, 12, 13, 14, 15]$.

### D.3. Image generation

**Our method.** We use the official publicly released code from ReDi (Kouzelis et al., 2025b) with a single modification: we replaced PCA with our VAE. The number of VAE channels matches the PCA rank. Additionally, our VAE compresses the same DINO features as PCA. Specifically, for a fair comparison with ReDi, we avoid the multi-scale feature space and use only final layer features.

**Training.** We use the official publicly released code from ReDi (Kouzelis et al., 2025b), removing gradient checkpointing since our hardware supported training without it. All hyperparameters remain identical to those in the original paper. However, we use different hardware: 4 NVIDIA GH200 Grace Hopper Superchips instead of 8 A100 (40GB) GPUs.

**Sampling.** We follow the same sampling procedure as ReDi (Kouzelis et al., 2025b), with official hyperparameters from the paper.

**Evaluation.** We follow the evaluation approach from ReDi (Kouzelis et al., 2025b). Since the paper does not report SiT-B or SiT-XL results with representation guidance at 400K training steps, we trained both models from scratch using the official PCA checkpoint and training protocol, incorporating the modifications described above. We performed a hyperparameter sweep over representation guidance strength $w_r \in \{1.1, 1.2, 1.5\}$ (values from the original paper) for all methods and reported the best results.

## E. Limitations and Future Work

Current limitations include (i) higher sampling latency compared to single-shot regression, (ii) mild long-horizon chroma drift, and (iii) reliance on upstream VFM domain coverage; moreover, achieving state-of-the-art video quality was not a primary objective.

Looking ahead, we plan to investigate several directions: (i) factorized video generation, i.e., training diffusion models directly in the latent space of video-centric VFMs' VAEs (Assran et al., 2025; Carreira et al., 2025) paired with lightweight RGB decoders, to improve computational efficiency and long-range stability; (ii) integrating DiffusionForcing(Chen et al., 2025a) to sustain high-fidelity predictions over extended sequences and (iii) designing a domain-specific diffusion architecture incorporating causal VAE components (Wan et al., 2025).

