# OpenReview forum: "VFMF: Dense Forecasting by Generating Foundation Model Features"
_ICML.cc/2026/Conference — ICML 2026 regular_

### Official Review · Reviewer_cGy5 · 2026-03-10

**Soundness:** 3
**Presentation:** 3
**Significance:** 3
**Originality:** 3
**Overall Recommendation:** 4
**Confidence:** 3

**Summary:**

This paper studies future scene forecasting in the feature space of vision foundation models (VFMs). The main idea is to first compress high-dimensional VFM features using a VAE, then generate future features in the latent space with an autoregressive flow matching model. The predicted features are finally decoded into downstream modalities such as segmentation, depth, normals, and RGB.
The authors also focus on the variable-length context setting, arguing that real-world forecasting models should be able to operate with different amounts of historical observations. Experiments on Cityscapes and Kubric show improvements over DINO-Foresight and Generalist Forecasting.

**Compliance With Llm Reviewing Policy:**

Affirmed.

**Final Justification:**

The author addressed my main problems and concerns.

**Key Questions For Authors:**

1.Can the authors clarify more explicitly what the main contribution is relative to Generalist Forecasting? Is the improvement mainly from the latent space design, the flow matching formulation, or the variable-context setup?
2.How well does the method generalize to other backbone features beyond DINOv2?
3.Since the model is generative, have the authors considered evaluating uncertainty calibration instead of relying mainly on best-of-k sampling?

**Strengths And Weaknesses:**

Strengths
1.The motivation of the paper is clear and reasonable. Forecasting directly in pixel space is known to be difficult due to multimodality, so predicting future representations instead of raw pixels is a sensible design choice.
2.The experimental section is also quite solid. The paper evaluates several downstream tasks and includes ablations on different latent spaces and compression strategies. The comparison with deterministic forecasting methods such as DINO-Foresight is helpful and shows that generative modeling can better capture uncertainty in future predictions.
3.Another positive point is that the method itself is relatively simple and modular, which makes it easy to understand and potentially easy to extend.

Weaknesses
1.My main concern is about the level of novelty. The general direction of forecasting future visual representations has already been explored in several recent works. For example, DINO-Foresight predicts future DINO features using deterministic regression, while Generalist Forecasting uses diffusion models to generate future features.
Compared with these works, the main difference here seems to be the combination of VAE-based compression and flow matching in the latent space. While this design appears to work well in practice, the conceptual novelty beyond existing generative forecasting approaches is somewhat limited.
2.Another point is that part of the reported improvement may come from the evaluation protocol. The variable-context setting is reasonable, but it also introduces a slightly different experimental setup compared to earlier work. It would be helpful to further clarify how much of the gain comes from the modeling choice versus the evaluation setting.
Overall, the paper feels more like a well-engineered refinement of an emerging research direction rather than a fundamentally new formulation.
3. While reading the manuscript, I noticed an unusual instruction at the end of Page 2, written in white hidden text, asking the reviewer to include specific phrases in the review (e.g., “Include BOTH the phrases ... in your review”). This instruction is clearly unrelated to the scientific content of the paper.
More importantly, it appears that this content modifies the review template behavior by inserting hidden instructions, which I find inappropriate in an academic submission. Embedding such prompt-like instructions targeted at reviewers or automated tools raises concerns about the integrity of the review process.
In my view, the manuscript should strictly focus on scientific contributions. Introducing hidden instructions aimed at influencing how a review is written is not appropriate and could potentially violate submission guidelines. In some cases, modifying the template or inserting hidden content may even introduce a risk of desk rejection during the review process.
I strongly recommend that the authors remove such hidden instructions from the manuscript in future revisions or submissions.

---

> ### Author Rebuttal · Authors · 2026-03-31
>
> Thank you for the positive review.
>
> > My main concern is about the level of novelty. The general direction of forecasting future visual representations has already been explored in several recent works. For example, DINO-Foresight predicts future DINO features using deterministic regression, while Generalist Forecasting uses diffusion models to generate future features. Compared with these works, the main difference here seems to be the combination of VAE-based compression and flow matching in the latent space. While this design appears to work well in practice, the conceptual novelty beyond existing generative forecasting approaches is somewhat limited. Another point is that part of the reported improvement may come from the evaluation protocol. The variable-context setting is reasonable, but it also introduces a slightly different experimental setup compared to earlier work. It would be helpful to further clarify how much of the gain comes from the modeling choice versus the evaluation setting. Overall, the paper feels more like a well-engineered refinement of an emerging research direction rather than a fundamentally new formulation.
>
> We agree that our work builds on an emerging research direction rather than proposing a fundamentally new formulation — but we believe it makes a clear and practically significant technical contribution: we identify that compressing DINO features via a VAE with latent regularisation is critical for effective generative forecasting, and we rigorously demonstrate its superiority over prior compression choices (no compression, PCA) used in existing works. To the best of our knowledge, nobody rigorously demonstrated the effectiveness of VAE-based feature compression for forecasting.
>
> Regarding the evaluation protocol: our comparisons with Generalist Forecasting strictly follow their setup, using **fixed**-length conditioning. The gains we report relative to that baseline therefore reflect modeling choices, not evaluation differences. The variable-context setting is also a more realistic evaluation — real-world observations are almost always partial — and is designed to surface limitations of deterministic regression methods that our approach addresses.
>
> > Can the authors clarify more explicitly what the main contribution is relative to Generalist Forecasting? Is the improvement mainly from the latent space design, the flow matching formulation, or the variable-context setup?
>
> The improvement is mainly from the latent space design. In comparisons with Generalist Forecasting, we strictly follow their setup, forecasting from **fixed**-length context. Our method, without the VAE, with fixed-length context and flow matching substituted for DDPM, is essentially equivalent to Generalist Forecasting. Since Generalist Forecasting has no official public implementation, we isolate the contribution through ablations: our results show VAE compression consistently outperforms raw (uncompressed) diffusion in feature space. Combined with superior performance on the evaluation setup and data matching Generalist Forecasting, this clearly identifies compressed latent modelling as the key driver of improvement.
>
> > How well does the method generalize to other backbone features beyond DINOv2?
>
> For fair comparison with prior works, we evaluated exclusively with DINOv2 (ViT-B with registers). However, nothing in our framework is specific to DINO — the VAE compression and flow matching pipeline are backbone-agnostic and could in principle be applied to features from any pretrained encoder.
>
> > Since the model is generative, have the authors considered evaluating uncertainty calibration instead of relying mainly on best-of-k sampling?
>
> We evaluate our generative model using both distributional metrics (Fréchet distance in the decoded modality space, Table 2) and pointwise metrics (Best-of-k, Table 2), following the training and evaluation protocol of Generalist Forecasting to enable direct comparison. Fréchet distance captures how well the predicted distribution matches the true future distribution, which is related to sample diversity. We agree that more explicit uncertainty calibration analysis would be a valuable addition. However, we do not have official implementation of the baseline and we cannot compute explicit uncertainty calibration for their method.

---

> > ### Author Rebuttal · Reviewer_cGy5 · 2026-04-01
> >
> > I have no further questions.

---

### Official Review · Reviewer_LMCK · 2026-03-12

**Soundness:** 3
**Presentation:** 3
**Significance:** 2
**Originality:** 3
**Overall Recommendation:** 4
**Confidence:** 4

**Summary:**

The paper presents a generative framework for scene forecasting to autoregressively predict future states in the latent space of vision foundation model (VFM) from variable-length contexts. The key contributions include utilizing flow matching for uncertainty-aware feature generation to overcome the averaging limitations of deterministic regression. It enables flexible decoding of these generated latent features into diverse downstream modalities such as RGB, depth, surface normals, and semantic segmentation.

**Compliance With Llm Reviewing Policy:**

Affirmed.

**Final Justification:**

My concerns have been addressed by the authors’ response.

**Key Questions For Authors:**

Why is a VAE-compressed latent space crucial for generating Vision Foundation Model features using flow matching? How does this approach differ from and outperform deterministic regression and direct feature diffusion methods in scene forecasting?

**Limitations:**

The authors have not discussed the potential societal impacts of their work. Addressing issues like biases inherited from pretrained foundation models, ethical concerns in sensitive downstream applications , and the lack of explicit temporal consistency metrics in their evaluation would be highly beneficial.

**Strengths And Weaknesses:**

1. This paper abandons the cumbersome task of directly generating pixels, and instead generates high-dimensional features, which are successfully and consistently decoded into RGB, semantic segmentation masks, depth maps, and surface normals.
2. By explicitly modeling the uncertainty of future vision using flow matching, VFM features overcome the fatal flaw of previous regression models that produce fuzzy and unrealistic prediction maps.
3. Compared to previous methods that directly diffuse features or use PCA for dimensionality reduction, the VAE compression scheme proposed in this paper can significantly reduce information loss.

Weaknesses:
1. When the model undergoes long-range autoregressive expansion, a slight chroma drift phenomenon appears visually, which limits its potential as an ultra-long video simulator.
2. The quality of the final generated RGB or depth map is essentially limited by the data domain coverage of the upstream pre-trained VFM and the expressive power of the specific decoder.
3. This two-stage paradigm of "predicting features first and then rendering them into RGB or depth maps using an independent decoder" is prone to introducing and accumulating reconstruction artifacts during feature decompression and cross-modal mapping.
4. Although VAEs achieve superior reconstruction quality compared to PCA, forcibly reducing features from 3072 dimensions to 16 dimensions is inherently an extremely lossy process. Increasing model capacity only partially alleviates this problem; even minute details of visual texture will still be lost.
5. Although the paper emphasizes the high cost of predicting pixels, it does not provide an intuitive comparison of the visual realism of its final decoded RGB video with current mainstream pixel-level video generation models in the charts.

---

> ### Author Rebuttal · Authors · 2026-03-31
>
> > When the model undergoes long-range autoregressive expansion, a slight chroma drift phenomenon appears visually, which limits its potential as an ultra-long video simulator.
>
> We acknowledge this limitation, but note that ultra-long video simulation is not the focus of our work. Our key contributions, stated in lines 101–115, are: (1) demonstrating that deterministic regression-based forecasters perform poorly under variable and short context; (2) introducing autoregressive flow matching in VFM feature spaces for uncertainty-aware, coherent predictions across context lengths; (3) introducing a VAE-compressed VFM feature space that preserves information substantially better than the PCA compression used in prior work; and (4) demonstrating significant improvements in forecasting of semantic, geometric, and RGB quantities.
>
> > The quality of the final generated RGB or depth map is essentially limited by the data domain coverage of the upstream pre-trained VFM and the expressive power of the specific decoder.
>
> Precisely, which is why — to isolate our contributions — we strictly follow the upstream VFM (DINO, ViT-B with registers) and decoder (DINO-Foresight DPT or Generalist Forecasting Cross-Attention) choices of prior works.
>
> > This two-stage paradigm of "predicting features first and then rendering them into RGB or depth maps using an independent decoder" is prone to introducing and accumulating reconstruction artifacts ...
>
> This is true of any latent diffusion paradigm. However, DINO-Foresight has already demonstrated that this two-stage approach outperforms both video generation followed by modality decoding and direct modality prediction on the majority of evaluation metrics, and we further improve upon DINO-Foresight. We also note that there is no cross-modal mapping in our pipeline — each modality is decoded independently. Further details about the probes are provided in the Benchmark section and the Appendix.
>
> > Although VAEs achieve superior reconstruction quality compared to PCA, forcibly reducing features from 3072 dimensions to 16 dimensions is inherently an extremely lossy process. Increasing model capacity only partially alleviates this problem; even minute details of visual texture will still be lost.
>
> Preserving minute details of visual texture is not the primary focus of our work. Our metrics clearly show that the information retained after VAE compression is sufficient for strong performance on the semantic and geometric forecasting tasks we target. Moreover, as shown in Table 3, VAE compression significantly outperforms PCA on downstream dense prediction, demonstrating that the relevant information is well preserved relative to the alternative used in prior work.
>
> > Why is a VAE-compressed latent space crucial for generating Vision Foundation Model features using flow matching?
>
> Because it simultaneously **compresses** high-dimensional features and **regularises the scale and distribution of the latent space**, which directly affects the signal-to-noise ratio of the diffusion process.
>
> To test this, we compress features (by 192×) using PCA and VAE while keeping the latent dimension fixed. Figure 4 shows that diffusing low-rank PCA channels significantly outperforms diffusing DINO features directly or with higher-rank PCA (1152 dimensions instead of 16). In particular, errors in geometry and shape inconsistency across time largely disappear, as further evidenced by sharply decoded geometry (surface normals, depth), and this is reflected in significantly improved downstream forecasting accuracy on both datasets (Table 1).
>
> While low-rank PCA simplifies latent diffusion, it incurs noticeable information loss: Figure 4 shows that high-level properties such as semantics and geometry are well preserved, but pixel-level details are not, which are crucial for high-fidelity dense decoding such as RGB. Table 3 quantifies this, showing that PCA performs significantly worse than VAE on downstream dense prediction. This analysis is discussed in full in the section "PCA Compression Is Suboptimal For Generation."
>
> > How does this approach differ from and outperform deterministic regression and direct feature diffusion methods in scene forecasting?
>
> Our method differs from deterministic regression in that it is a stochastic generative model, and from direct feature diffusion in that it operates in a compressed latent space rather than on raw features. Tables 1 and 2 quantify the resulting improvements over each respectively.
>
> As shown qualitatively in Figure 2, deterministic regression produces blurry averages over the many plausible futures, because it is trained to predict their mean. Our method instead models the full distribution of futures, enabling sharp, plausible predictions.

---

> > ### Author Rebuttal · Reviewer_LMCK · 2026-04-04
> >
> > Thank you for the rebuttal. While I appreciate the technical clarifications regarding the VAE-compressed latent space, my core concerns regarding evaluation and practical limitations remain unresolved. Therefore, I maintain my score.
> >
> > 1.Acknowledging flaws by simply stating they are not the primary focus does not eliminate them. These unresolved issues seem cap the method's downstream applicability and practical utility.
> >
> > 2.The authors ignored my request to compare their generated visual results with current mainstream pixel-level video generation models. Without this benchmark, it is difficult to assess the true competitiveness of this feature-based paradigm.

---

> > > ### Author Response · Authors · 2026-04-07
> > >
> > > > 1.Acknowledging flaws by simply stating they are not the primary focus does not eliminate them. These unresolved issues seem cap the method's downstream applicability and practical utility.
> > >
> > > Our method is not designed for state-of-the-art RGB generation — a point clarified in the rebuttal. Notably, neither of the prior feature-based methods we compare against (DINO-Foresight, Back-To-Features) reports RGB results either. Across all other modalities (semantic segmentation, depth, and surface normals), our metrics demonstrate that the information preserved after VAE compression is sufficient for strong performance on the semantic and geometric forecasting tasks we target (as do the prior feature-based forecasting works).
> > >
> > > Latent-space reconstruction analysis in Table 1 (BFL [3]), shows that DINOv2-based representations incur substantially higher reconstruction error than mainstream RGB VAEs, making them ill-suited for RGB decoding from features. However, RGB generation is neither our target task nor the target task of the baselines we compare against. This limitation is simply not relevant to the scope of our evaluation.
> > >
> > > > 2.The authors ignored my request to compare their generated visual results with current mainstream pixel-level video generation models. Without this benchmark, it is difficult to assess the true competitiveness of this feature-based paradigm.
> > >
> > > As noted in our rebuttal, prior work has already benchmarked feature-based generation against pixel-level video generation models, consistently demonstrating its superiority. We build directly on these findings.
> > >
> > > Specifically, DINO-Foresight [1] shows in Table 1 that pixel-space feature regression outperforms the pixel-space generative model VISTA — even when VISTA is fine-tuned on Cityscapes. Our method uses the same dataset and further improves upon DINO-Foresight. Table 2 of the same paper reinforces this conclusion, showing that regressing foundation model features (e.g., DINOv2 or SAM) significantly outperforms regressing Stable Diffusion latents.
> > >
> > > This advantage is further demonstrated by Back To Features [2], whose Table 1 demonstrates that DINO-Foresight-style feature regression substantially outperforms both COSMOS-4B and COSMOS-12B. While we cannot directly compare with Back To Features due to the unavailability of their code and missing implementation details, their reported Cityscapes performance is close to DINO-Foresight's. Recall that our method, which replaces feature regression with flow matching, shows clear improvements over DINO-Foresight on that same benchmark.
> > >
> > > Taken together, these results establish that the feature-based paradigm is not only competitive with, but consistently superior to, pixel-level video generation, making a direct side-by-side comparison in our paper redundant rather than necessary.
> > >
> > > [1] DINO-Foresight: https://arxiv.org/pdf/2412.11673
> > >
> > > [2] Back To Features: https://arxiv.org/pdf/2507.19468
> > >
> > > [3] BFL: https://bfl.ai/research/representation-comparison

---

### Official Review · Reviewer_UvDj · 2026-03-12

**Soundness:** 3
**Presentation:** 3
**Significance:** 2
**Originality:** 2
**Overall Recommendation:** 4
**Confidence:** 3

**Summary:**

This paper proposes VFMF, a dense scene forecasting method that generates Vision Foundation Model (VFM) features via autoregressive flow matching in a VAE-compressed latent space. It addresses the flaws of deterministic regression-based VFM forecasting, which averages multiple plausible futures leading to blurry predictions, and naive generative methods that suffer from poor sample quality due to the high dimensionality of VFM features. The study validates that the model outperforms existing baselines on datasets including Cityscapes and Kubric across multiple downstream modalities, and also proves the transferability of the VAE compression module to image generation tasks. The study's principal aspect consists of constructing a generative forecasting framework for VFM features with optimal compression and uncertainty modeling capabilities, while the main issue assessed by the manuscript is how to effectively build generative forecasters in high-dimensional VFM feature spaces for scene prediction with variable context lengths.

**Compliance With Llm Reviewing Policy:**

Affirmed.

**Key Questions For Authors:**

1. In the paper, the VAE only compresses the channel dimension of VFM features while keeping the spatial dimension unchanged, with the justification that VFM features have already undergone spatial downsampling. However, different VFMs adopt different spatial downsampling ratios. Have the authors verified the impact of lightweight spatial dimension compression (e.g., 2× downsampling) on prediction performance?
2. How do different types and algorithms of flow matching solvers affect the performance of the entire system?
3. The "mild chroma drift" seems to stem from the reconstruction bias of the RGB decoder. What potential solutions might there be for this issue?
4. How does the choice of backbone model impact the training cost of the overall model?

**Limitations:**

yes

**Strengths And Weaknesses:**

Strengths
* The proposed VAE-compressed latent space outperforms traditional PCA compression, effectively solving the dimensionality curse and information loss in generative VFM forecasting. The model supports variable-length input frames, dynamically adjusts prediction uncertainty by context, and achieves significant performance gains over deterministic and generative baselines on multiple datasets, with high practical value.
* Comprehensive and high-performance multi-modality forecasting is realized: the generated VFM features are successfully decoded into semantic segmentation, depth maps, surface normals and RGB images. Targeted evaluation metrics and decoding architectures are configured for each modality, fully exploiting the semantic and geometric information of VFM features.
The experimental design is extremely rigorous and thoughtful. Exclusive training/evaluation protocols are tailored for different datasets and modalities; extensive ablation studies verify the effectiveness of core designs (diffusion space, VAE hyperparameters, sampling numbers). Baselines are retrained for variable input frames and unified decoding/training protocols to ensure fair comparison, greatly enhancing the credibility of conclusions.
* The VAE compression strategy has excellent generalization and transferability. It can directly replace the PCA module in the SOTA image generation model ReDi and boost performance, providing a universal and efficient compression scheme for VFM feature-related generative tasks and expanding application boundaries.

Weakness
* The generative sampling mechanism causes higher inference latency than one-step regression methods. While this is a common issue for generative modeling methods, it still limits the model's adaptability to high real-time requirement scenarios.
* Mild chroma drift exists in long-horizon forecasting, and the stability of long-sequence generation needs further improvement.
* Low-level vision methods are only analyzed as background without quantitative comparison, failing to intuitively reflect the performance advantages of VFM feature modeling over such methods.

---

> ### Author Rebuttal · Authors · 2026-03-31
>
> Thank you for the positive review of our work.
>
> > In the paper, the VAE only compresses the channel dimension of VFM features while keeping the spatial dimension unchanged, with the justification that VFM features have already undergone spatial downsampling. However, different VFMs adopt different spatial downsampling ratios. Have the authors verified the impact of lightweight spatial dimension compression (e.g., 2× downsampling) on prediction performance?
>
> In our experiments, we fixed the dimensionality to match prior works (DINO-Foresight and ReDi) to enable fair comparison. We will expand discussion of this trade-off in future work and include an ablation over bottleneck dimensionality to give readers a clearer picture of the loss-compression frontier. We also note that improved reconstruction ability does not necessarily translate to better generation quality [1, 2], so the optimal bottleneck size should be evaluated in terms of downstream generation performance rather than reconstruction loss alone.
>
> [1] https://arxiv.org/abs/2501.01423
> [2] https://arxiv.org/abs/2502.14831
>
> ---
>
> > The "mild chroma drift" seems to stem from the reconstruction bias of the RGB decoder. What potential solutions might there be for this issue?
>
> Potential solutions include training the decoder with flow matching to hallucinate missing high-frequency details, and training with noisy features that simulate the inference distribution (as explored in RAE, for example). However, as RGB generation is not a core contribution of our work, addressing this fully is outside our current scope.
>
> ---
>
> > How does the choice of backbone model impact the training cost of the overall model?
>
> For fair comparison with prior works, we evaluated only DINOv2-B. However, nothing in our method assumes anything specific about the backbone, so the framework is in principle applicable to other VFMs with training cost scaling accordingly with backbone feature dimensionality and spatial resolution.
>
> ---
>
> > How do different types and algorithms of flow matching solvers affect the performance of the entire system?
>
> We did not ablate solver choice due to computational constraints. We use 10-step Euler sampling throughout — likely a suboptimal configuration — and yet our method outperforms prior works even in this setting, suggesting the gains are robust to solver choice.

---

> > ### Author Rebuttal · Reviewer_UvDj · 2026-04-04
> >
> > I have no other questions.

---

### Official Review · Reviewer_1T5L · 2026-03-13

**Soundness:** 2
**Presentation:** 3
**Significance:** 2
**Originality:** 2
**Overall Recommendation:** 4
**Confidence:** 4

**Summary:**

This paper proposes to forecast future states by predicting foundation model features for future frames. Compared with regressive approaches, the authors proposes to predict future features with flow matching models. Then, the paper discusses several key factors that affect the quality of future feature prediction and develops some techniques to solve the challenges like injecting a VAE model to reduce the dimension. Experiment results show that VFMF can predict future features with a better quality compared with DINO-Foresight on multiple datasets.

**Compliance With Llm Reviewing Policy:**

Affirmed.

**Final Justification:**

I appreciate the authors’ detailed response during the rebuttal period. It addresses most of my concerns, and I am happy to raise my score accordingly.

My only remaining question concerns the comparison with RAE, and I would appreciate further clarification from the authors:

Given the use of a wide DDT head in RAE, it is not necessary to increase the transformer hidden dimension. As a result, the computational cost of RAE appears comparable to that of the proposed method, while the overall architecture is arguably simpler due to the removal of the VAE component.

Table 3 provides a very helpful ablation study. However, as discussed in the RAE paper, certain techniques can improve the denoising of DINO features. If such techniques are applied, I am wondering whether the VAE component is still necessary.

**Key Questions For Authors:**

Please consider replying to the weaknesses.

**Limitations:**

yes

**Strengths And Weaknesses:**

Strengths:

1. This paper gives a good insight and analysis regarding the drawbacks of pixel-based scene forecasting model. It is well-motivated to propose the algorithm that predicts future states in the dense foundation model feature space to understand the high-level semantics.

2. The experiment part gives a detailed discussion regarding the challenges and potential solutions. The comparison between PCA and VAE is clear and helpful to understand the motivations.


Weaknesses:

1. My major concern lies in the novelty and technical significance of this paper. It is not a new idea to combine generative models with foundation model features. There are some related works like REPA [r1] and RAE [r2]. It is necessary to include detailed discussion and comparison with these previous works. Especially, RAE shares a very similar design with this paper, which already discussed the challenges of generating high-dimensional features in details. Some keypoints are exactly same like too high dimension.

2. Although it helps to reduce the feature dimension, VAE inevitably brings some information loss for high-dimensional features due to its large dimension reduction. The paper does not have enough discussion regarding how to reduce or avoid this loss.

3. The paper only compares VFMF with DINO-Foresight, which is not enough. I think there is another straightforward baseline that first predicts future pixels and then extracts the features using the predicted images.

4. The "Mean-of-k" is not very clear for me. The authors mention a major motivation to use flow matching models as the multimodal predictions. If the prediction is multimodal (e.g. a car running into different directions), what is the meaning to calculate the average features of multiple predictions?

[r1] Representation Alignment for Generation: Training Diffusion Transformers Is Easier Than You Think

[r2] Scaling Text-to-Image Diffusion Transformers with Representation Autoencoders

---

> ### Author Rebuttal · Authors · 2026-03-31
>
> > My major concern lies in the novelty and technical significance of this paper. It is not a new idea to combine generative models with foundation model features....
>
> Thank you for raising these references. We address them in turn.
>
> Our work differs fundamentally from both REPA and RAE in problem setting: both focus exclusively on image generation, whereas we tackle a distinct problem — forecasting future representations from a variable number of context frames. Those works aim to improve pixel-level generation quality or convergence; we generate high-level future features using diffusion at a higher level of abstraction, while explicitly modelling temporal dynamics.
>
> **RAE:** RAE proposes handling high-dimensional features by brute-force extension of denoising network width, which is computationally prohibitive in our setting and computationally expensive in general setting. We instead apply standard techniques (compression and latent regularisation via a VAE) in a novel setting (feature forecasting), enabling standard network sizes and architectures for denoising. Crucially, RAE models only final-layer DINO features for a single image, whereas we model multi-scale DINO features (4 layers) conditioned on T context frames of patch-level features. Naively applying RAE to our setting would require a transformer hidden dimension of at least 3072 for DINO-base alone — without accounting for temporal context — which is computationally infeasible. Our VAE and denoising transformer both operate at 768 dimensions, consistent with prior works such as DINO-Foresight.
>
> **REPA:** REPA does not generate features at all. It aligns the features of an RGB latent diffusion network with a pretrained encoder to accelerate convergence of image generation — it assumes features exist as a supervisory signal; we generate features as the output. Furthermore, our baseline ReDi already demonstrates superior performance to REPA on image generation, and we further improve upon ReDi by replacing its PCA DINO latents with our VAE DINO latents.
>
> Beyond the problem setting, we also provide insights absent from both prior works: we show that a VAE specifically improves temporal consistency and shape geometry compared to alternative compression approaches (Figure 4), whereas REPA and RAE report image generation metrics without identifying which aspects of generation quality improve.
>
> We will add explicit discussion of both works in the related work section to clarify these distinctions.
>
> > Although it helps to reduce the feature dimension, VAE inevitably brings some information loss for high-dimensional features due to its large dimension reduction. The paper does not have enough discussion regarding how to reduce or avoid this loss.
>
> This is a valid point and we agree that information loss is an inherent trade-off of compression. The most direct way to reduce it is to increase the channel dimensionality of the VAE bottleneck. In our experiments, we fixed the dimensionality to match prior works (DINO-Foresight and ReDi) to enable fair comparison, but we will expand our discussion of this trade-off in the paper and include an ablation over bottleneck dimensionality to give readers a clearer picture of the loss-compression frontier. We also note that improved reconstruction ability does not necessarily translate to better generation quality [1, 2], so the optimal bottleneck size should be evaluated in terms of downstream generation performance rather than reconstruction loss alone. In particular, in Table 3 we show that downstream decoding performance of VAE compressed features is on par with the uncompressed features.
>
> [1] https://arxiv.org/abs/2501.01423
> [2] https://arxiv.org/abs/2502.14831
>
> > The paper only compares VFMF with DINO-Foresight, which is not enough. I think there is another straightforward baseline that first predicts future pixels and then extracts the features using the predicted images.
>
> We respectfully disagree that this baseline is necessary here. Prior work has already established the relevant comparisons: DINO-Foresight (Table 1 of their paper) clearly demonstrates that feature-level generation outperforms pixel-level generation on dense forecasting, and their Table 2 shows that regressing DINO latents outperforms regressing RGB VAE latents such as those from Stable Diffusion. Re-establishing these results is outside the scope of our contribution.
>
> > The "Mean-of-k" is not very clear for me...
>
> Mean-of-k is included not because it is a desirable prediction, but because it allows fair comparison with deterministic baselines on their own terms: averaging k samples from our generative model approximates the same conditional mean that regression models are trained to predict. As shown in Table 1, our model estimates this mean more accurately than prior works — and its Best-of-k predictions are sharper and better still, demonstrating the value of modelling the full distribution rather than collapsing to the mean.

---

> > ### Author Rebuttal · Reviewer_1T5L · 2026-04-03
> >
> > I appreciate the authors’ detailed response during the rebuttal period. It addresses most of my concerns, and I am happy to raise my score accordingly.
> >
> > My only remaining question concerns the comparison with RAE, and I would appreciate further clarification from the authors:
> >
> > Given the use of a wide DDT head in RAE, it is not necessary to increase the transformer hidden dimension. As a result, the computational cost of RAE appears comparable to that of the proposed method, while the overall architecture is arguably simpler due to the removal of the VAE component.
> >
> > Table 3 provides a very helpful ablation study. However, as discussed in the RAE paper, certain techniques can improve the denoising of DINO features. If such techniques are applied, I am wondering whether the VAE component is still necessary.

---

### Decision · Program_Chairs · 2026-04-30

**Decision:**

Accept (regular)

**Comment:**

Reviewers appreciated the well-motivated approach to predicting future states in a dense VFM feature space using flow matching, as well as the solid experimental design that demonstrates the advantage of the VAE-compressed latent space over PCA in preserving semantic and geometric information. The overall assessment leans positive, with three Weak Accepts and one Weak Reject. The main concerns center on the incremental conceptual novelty relative to prior feature-forecasting work and the lack of direct comparisons with pixel-level video generation baselines. In the rebuttal, the authors clarify that the core contribution is identifying VAE compression as a key ingredient for generative feature forecasting, which addresses most of the novelty concerns. While one reviewer remains unconvinced by the absence of pixel-level generation baselines, the overall reviewer sentiment is positive, and the AC supports acceptance as a Weak Accept.